# The aging mouse CNS is protected by an autophagy-dependent microglia population promoted by IL-34

Rasmus Berglund [1] ✉, Yufei Cheng[1], Eliane Piket[1], Milena Z. Adzemovic[1], Manuel Zeitelhofer [2], Tomas Olsson[1,3], Andre Ortlieb Guerreiro-Cacais [1,3] & Maja Jagodic [1,3]

Microglia harness an unutilized health-promoting potential in age-related neurodegenerative and neuroinflammatory diseases, conditions like progressive multiple sclerosis (MS). Our research unveils an microglia population emerging in the cortical brain regions of aging mice, marked by ERK1/2, Akt, and AMPK phosphorylation patterns and a transcriptome indicative of activated autophagy - a process critical for cellular adaptability. By deleting the core autophagy gene *Ulk1* in microglia, we reduce this population in the central nervous system of aged mice. Notably, this population is found dependent on IL-34, rather than CSF1, although both are ligands for CSF1R. When aging mice are exposed to autoimmune neuroinflammation, the loss of autophagy-dependent microglia leads to neural and glial cell death and increased mortality. Conversely, microglial expansion mediated by IL-34 exhibits a protective effect. These findings shed light on an autophagy-dependent neuroprotective microglia population as a potential target for treating age-related neuroinflammatory conditions, including progressive MS.

Microglia are resident myeloid cells of the central nervous system (CNS). They act both as glial cells, preserving CNS homeostasis with trophic support to neurons and other glia, and as immune cells carrying functions important in response to tissue damage. While other glial cells are of ectodermal origin, the yolk sac-derived microglia populate the CNS during embryonic stages and are instrumental in neural development[1–3]. After establishment, microglia are considered self-renewing throughout life with region-specific density and phenotypes[4,5].

The establishment and maintenance of the microglial population is dependent on the activity of CSF1R[6–8]. In humans, mutations in the *CSF1R* gene cause severe loss of microglia and associate with lethal abnormalities and degenerative changes in the CNS[9–11]. In adult microglia depletion models, using either genetic targeting or chemical CSF1R inhibition, the niche is rapidly repopulated by bone marrow-derived macrophages (BMDM) and proliferation of residual microglia cells[11–13]. BMDM can enter the CNS during inflammation but does not contribute to the myeloid population of the CNS parenchyma when homeostasis is restored[14,15].

The CSF1R has two known ligands, colony-stimulating factor-1 (CSF-1/M-CSF) and interleukin-34 (IL-34). CSF-1 is highly expressed in the CNS by microglia/macrophages operating in an autocrine fashion, while IL-34 is primarily expressed by neurons and endothelial cells[16–18]. Neutralizing CSF-1 causes a partial loss of microglia primarily in the cerebellum and white matter regions of CNS, while anti-IL-34 antibody blocking causes a loss in grey matter, which follows the spatial expression patterns of these cytokines[7,19–21]. CSF-1 expression is dynamic and shown to increase during inflammation, while IL-34 has higher CSF1R affinity and a ~300-fold higher concentration in the CNS under homeostatis[22–24]. IL-34 concentrations in

[1]Department of Clinical Neuroscience, Division of Neuro, Karolinska Institutet, Center for Molecular Medicine, Karolinska University Hospital, 171 76 Stockholm, Sweden. [2]Department of Medical Biochemistry and Biophysics, Division of Vascular Biology, Karolinska Institutet, 171 65 Solna, Sweden. [3]These authors contributed equally: Tomas Olsson, Andre Ortlieb Guerreiro-Cacais, Maja Jagodic. ✉e-mail: rasmus.berglund@ki.se

cerebrospinal fluid do not however change during CNS inflammation[24].

Canonical-(macro)-autophagy is an intracellular process necessary for dynamic cellular adaptation to nutrient availability as well as intracellular metabolic, translational and organelle function disturbances, which impacts cell survival and differentiation. Downstream of the CSF1R activation, phosphorylation of ERK1/2, Akt and AMPK differentially regulate myeloid cell proliferation, survival, and differentiation[7,25,26]. The Akt pathway acts through mTORC1, leading to inhibition of autophagy and induction of cell-cycle progression in response to metabolic and immunological stimuli[27,28]. AMPK counteracts Akt/mTORC1 and stimulates autophagy and myeloid cell functions such as phagocytosis and secretion of IL-1b, IL-10 and other both proinflammatory and inhibitory cytokines[29–34]. While Akt and AMPK act in an opposing manner on autophagy initiation, activation of ERK1/2 is both regulated by and regulates autophagy and is shown e.g., to induce auto-phagosome maturation[35–38]. In comparison to CSF-1, IL-34 is shown to induce a higher degree of CSF1R-mediated activating phosphorylation of ERK1/2 in cultured microglial cells[24].

By altering cellular components for recycling, including surface receptors and metabolic organelles, autophagy appears as an integrative regulator of immune cells, hence a target for pharmaceutical immunomodulation[39]. In recent years, findings by us and others have expanded the function of autophagy and its associated proteins to other roles, including phagocytosis and MHC class II presentation[40–42]. The ULK1 protein is not associated with these noncanonical functions but is indispensable for the early steps of canonical-autophagy and a regulatory target through phosphorylation downstream of CSF1R[26,43–45]. Degradation of misfolded or aberrant proteins is a hallmark of ULK1-derived autophagy, and impairment in this process causes ER stress-activated unfolded protein response, which is associated with cellular senescence and apoptosis[46,47].

Altered microglial phenotypes have been implicated in multiple sclerosis (MS), which is an autoinflammatory disease of the CNS where immune cells destroy myelin and induce neuronal damage causing neurological deficits. Successful targeting of the adaptive immune system reduces inflammation and clinical relapses. However, later stages of MS disease present a more degenerative phenotype with age as the strongest risk factor and histopathological association to microglial phenotypes, including an increased density of autophagosomes[48–52]. CSF1R-mediated depletion of microglia in animal models accelerates progressive neuroinflammation, and several genetic risk alleles for human neurological diseases are assigned to microglial dysfunction[53]. ERK1/2 phosphorylation is predicted to induce the reactive "disease-associated microglia" (DAM) phenotype detected in several neurodegenerative and inflammatory diseases including MS[54–56]. These reactive microglial phenotypes display increased expression of, e.g., ITGAX (CD11c), TREM2, CLEC7A, CSF-1, MHC class II, and associated APC molecules[55,57–59]. P2RY12, CX3CR1 and CSF1R expression define the contrasting "homeostatic microglia". Autophagy and the pathways that regulate it, i.e. mTOR, AMPK, Akt, and ERK, are implicated in MS pathogenesis from pathway analysis of MS risk genes[60], including *ATG4D* that is involved in phagosome maturation and encoded by one of the MS risk loci[60].

Recent studies illustrate a spatial and transcriptional heterogeneity of microglia populations. Insights from humans and experimental animals deficient in microglia, suggest reshaping of microglial phenotypes, rather than depletion, as a therapeutic option for diseases such as progressive MS and inflammatory neurodegeneration[5,52,61]. In this study, we investigated how age-associated alterations in CSF-1- and IL-34-induced CSF1R signaling polarize microglial populations and the capacity of an autophagy-dependent microglia subpopulation to control and dampen neuroinflammation.

## Results

### The aged CNS is characterized by a reduced turnover of microglia and an altered CSF1R axis

The turnover of microglia is slow compared to other immune cells. In human CNS, about 28% of the microglia population is replaced every year, and in rodents, less than 1% are newly divided cells at any given time[4,62]. The microglial proliferative rate shows regional differences, but the cells seem to proliferate locally without the contribution from distant migrating microglia[5]. To study the turnover of microglia during aging, we isolated microglia from aged CNS (>20 months) and compared them to microglia from adult mice (3-5 months). In these experimental settings, we used the established gating strategy defining microglia as live CD11b[+], Ly6G[-], CD45[Intermediate] or, in naïve conditions, by the *Cx3cr1* promoter-driven eYFP reporter expression (Fig. S1A, B). The CD45[Intermediate] population has lower expression of the monocyte marker Ly6C and increased expression of CD11c and CX3CR1(YFP), validating this gating strategy (Fig. S1A, B). We found decreased proliferation and apoptosis of aged microglia as quantified by EdU uptake (Fig. 1A) and AnnexinV staining (Fig. 1B), respectively. This did not affect the total number of microglia (Fig. 1C), suggesting a slower turnover of microglia in the aged CNS. Microglia are dependent on CSF1R signaling for proliferation and survival[20,23,63–66]. We found that the expression of CSF1R ligands, *Csf1* and *Il34*, was strongly upregulated in the aging CNS while CSF1R showed lower microglial surface expression (Fig. 1D, E). The aging microglia also displayed an altered surface marker profile, including increased expression of markers shaping the reactive microglia phenotype and function, namely MHC class II, CLEC7A, CD11c and TREM2 (Fig. 1F)[56–58,67,68]. In addition, the putative marker of homeostatic microglia, P2RY12 also showed increased density, while CX3CR1 was unchanged (Fig. 1F)[48,69,70]. Other receptors central in microglial maintenance and function, such as MSR1 (Macrophage scavenger receptor, SR-A) and CD68, showed no age-associated changes in expression (Fig. 1F). Taken together, this indicates both a higher degree of reactive differentiation in the aged microglia population with an associated increase in the homeostatic marker P2RY12.

Given the altered phenotype and balance in *Csf1/Il34*/CSF1R expression, we investigated CSF1R activated pathways in microglia ex vivo by quantifying phosphorylation of key molecules, Akt, ERK1/2 and AMPK, important in myeloid differentiation, survival and proliferation[7,26]. We found aged compared to adult microglia to have ex vivo significantly higher phosphorylation of the ERK1/2-Thr202/Thyr204 and the AMPK-Thr183/172 sites, while the phosphorylation of Akt-Ser473 (hereafter referred to as P-ERK1/2, P-AMPK and P-Akt, respectively) was reduced (Fig. 1G, H)[7]. These phosphorylations are known to activate several pathways downstream of CSF1R engagement[7,55,71]. Pharmacological inhibition of CSF1R signaling using PLX3397 abrogated phosphorylation of the P-ERK1/2 and P-AMPK sites demonstrating an engagement downstream of CSF1R (Fig. 1G, I)[72]. Furthermore, abrogation of ERK1/2 phosphorylation by the ERK1/2 inhibitor SCH772984 reduced microglial counts in cultures from aged CNS (Fig. 1J). This effect did not occur in cultures from young CNS (Fig. 1J), demonstrating an age-associated activation and dependency of this pathway in microglia.

In summary, we found aging microglia to have a reduced turnover and an altered regulatory response downstream of CSF1R in a CNS with age-related differences in expression of this receptor and its ligands. Activation of ERK1/2 and AMPK, with the deactivation of Akt, suggests increased canonical autophagy in the aging microglia population.

### The microglia of the aged CNS with reactive ERK1/2 pathway have a distinct transcriptional profile suggesting altered microglial functionality

To characterize the diverged populations, we sorted formaldehyde-fixed adult and aged microglia and further separated the aged

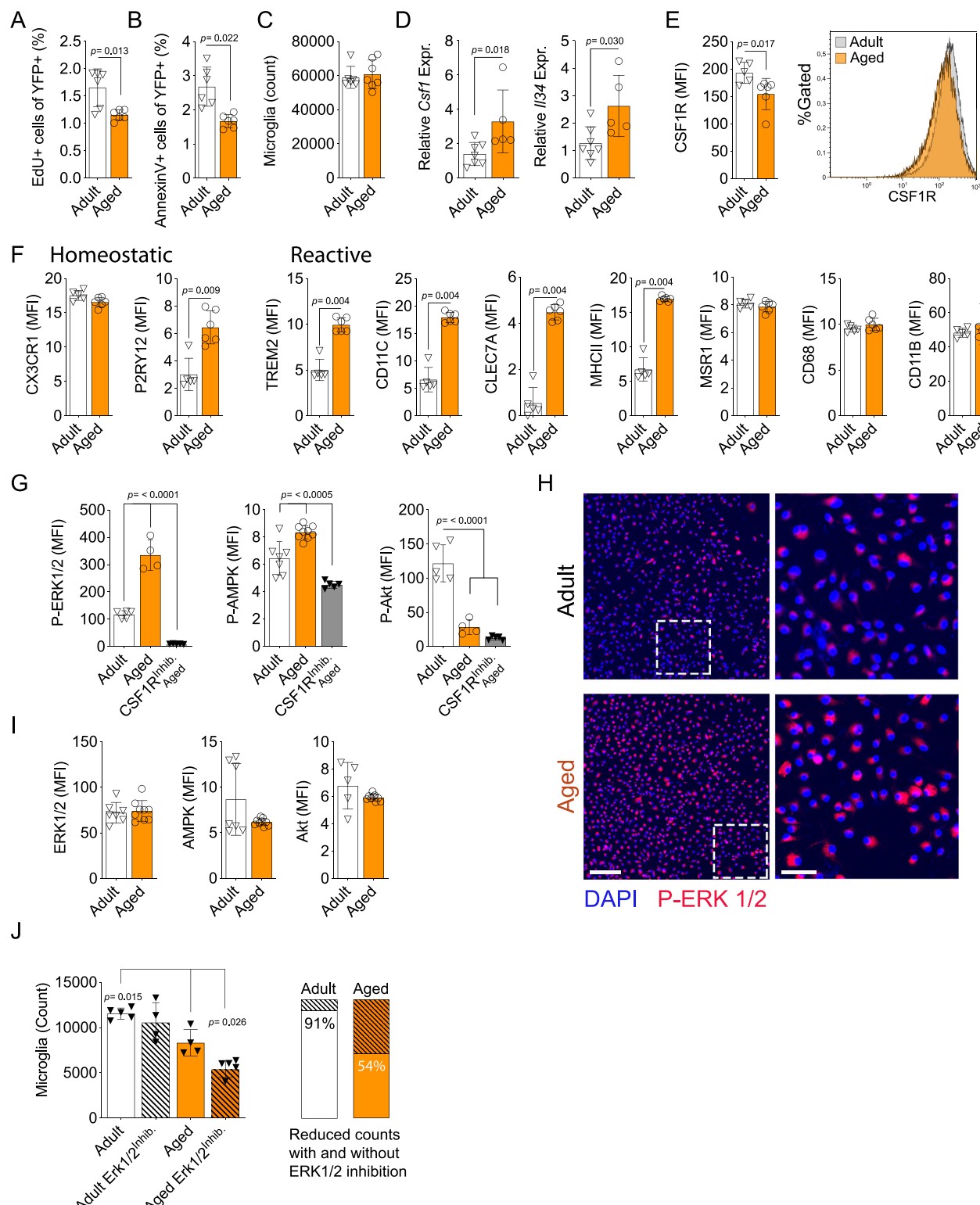

microglia based on their ERK1/2 phosphorylation state and analyzed the RNA expression of 770 neuroinflammation-related genes using Nanostring (Supplementary data 1, GEO accession number GSE248184). A heat map demonstrated a distinguished signature of the aged population with enhanced ERK1/2 phosphorylation, while the microglia from adult CNS and the aged P-ERK1/2$^{Low}$ microglia were similar (Fig. 2A, B). We detected 194 differentially expressed genes comparing P-ERK1/2$^{Low}$ and P-ERK1/2$^{High}$, while P-ERK1/2$^{Low}$ compared to adult CNS microglia only differed in 46 genes ($P < 0.01$, Log$_2$FC $> 1$)

(Fig. 2A, B, Supplementary data 1). Among most upregulated genes in the P-ERK1/2$^{High}$ microglia, we found *Rala*, a regulator of autophagy through activation of the ULK1 complex, necessary for canonical autophagy, as well as other key autophagy genes such as *Akt1*, *Atg3*, *Atg5*, *Atg9* and *Lamp1*. In line with this, gene enrichment analysis revealed a strong enrichment of the autophagy pathway in the aged P-ERK1/2$^{High}$ microglia (Fig. 2C). Phagocytosis and processing of debris are believed to be fundamental features of beneficial microglia function in several CNS pathologies and our transcriptome analysis

**Fig. 1 | Aged CNS is characterized by a reduced turnover of microglia and an altered CSF1R axis. A**−**C** Microglia proliferation, apoptosis and total numbers, respectively, assessed by flow cytometry after three EdU i.p. injections days 5, 3 and 1 before analysis (adult, $n = 5$; aged, $n = 6$). **D** CNS mRNA expression of *Csf1* and *Il34* normalized to *Hprt* and *B-actin* (adult $n = 7$; aged $n = 5$). **E, F** Microglia surface marker density assessed ex vivo by flow cytometry (adult $n = 5$; aged $n = 4$). **G** Microglia phosphorylation status assessed ex vivo after 4 h culture in CSF-1 enriched medium in the presence or absence of the CSF1R inhibitor PLX3397 (P-ERK1/2 and P-Akt adult, $n = 5$; aged, $n = 4$; CSF-1[Inhib], $n = 5$. P-AMPK; adult, $n = 7$; aged, $n = 9$; CSF-1[Inhib], $n = 5$). **H** Immunocytochemistry images of freshly isolated microglia of young and old mice showing expression of phosphorylated ERK (red) and nuclei (DAPI; blue). Scale bars correspond to 500 µm (left) and 50 µm (right). **I** Total intracellular content of target proteins in freshly isolated microglia assessed by flow cytometry PLX3397 (P-ERK1/2 and P-AMPK adult, $n = 7$; aged, $n = 9$. P-Akt; adult, $n = 5$; aged, $n = 9$; CSF-1R[Inhib], $n = 5$). **J** Microglia cell count after 72 h in vitro culture w/wo ERK1/2 inhibitor SCH772984. Bars to the right show percentual decrease in striped areas (all conditions $n = 5$ except "Adult CSF1R[Inhib] and Aged, $n = 4$). Adult mice were 3-5 months old, aged mice were >20 months old. **A**−**E** and **G** showing representative data from three independent experiments. **F, I** and **J** showing representative data from two independent experiments. **A**−**F** and **I** Mann-Whitney Two-tailed *U*-test. **G** and **J** One-way Anova followed by Tukeys post-hoc test. Error bars represent mean + SD. Source data are provided as a Source Data file.

associated ERK1/2 activated microglia to signatures found in human MS lesions described as "foamy" microglia (e.g., *Spp1, Msr1, Cd84*)[56]. We also found a substantial overlap to genes defined in the "Disease associated microglia" and TREM2 activated pathways detected in animal models of Alzheimer disease (Supplementary data 1). Among these genes are *Clec7a, Spp1, Cd68, Tlr2, Tlr7* and *Msr1*; all associated with reactive microglia and usually associated to inflammation and phagocytosis. However, the higher surface density of TREM2 detected (Fig. 1) was not supported by elevated mRNA expression in naïve state and the *Apoe* expression, associated with many reactive microglial phenotypes, was lower in the P-ERK1/2[High] population.

Genes associated with innate immune responses such as *Ccl2, Spp1, Cd74* and *Cd86* were strongly upregulated in the aged P-ERK1/2[High] microglia. Meanwhile, P-ERK1/2[Low] associated partly with homeostatic microglial signatures in MS and the AD animal models and had a higher expression of *Cd47* known to inhibit differentiation to a reactive macrophage phenotype (Fig. 2A, B, Supplementary data 1)[73,74]. The lipid metabolism pathway is believed to be of importance in macrophage biology as seen in e.g., "foamy" microglia found in MS lesions, and several of the represented genes are enriched in P-ERK1/2[High] population[56]. The enrichment of both autophagy- and phagocytosis-associated genes in the aged P-ERK1/2[High] microglia indicates a convergence of these pathways, which we have previously described after studying autophagy-associated phagocytosis in autoimmune neuroinflammation where we found myelin phagocytic function of CLEC7A-positive microglia to be necessary for recovery from experimental autoimmune encephalomyelitis (EAE)[75]. We validated this signature by detecting increased density of reactive microglial markers such as TREM2, MHC-II and CLEC7A in aged microglia (Fig. 1) which could be traced to the P-ERK1/2[High] population (Fig. 2D).

The gene expression pattern of the P-ERK1/2[High] population and elevated density of reactive markers thus indicate an activated canonical autophagy and microglial differentiation towards an immune activated state.

### ERK1/2-activated microglia show heightened autophagosome load and are vulnerable to autophagy deficiency

In order to elucidate the connection between ERK1/2 signaling and autophagy, we conducted ex vivo analysis detecting sustained presence of P-ERK1/2[High] microglia sorted from the aged CNS (Fig. 3A−C). Importantly, the P-ERK1/2[High] microglia from the aged CNS had higher autophagosome density, as defined by the presence of LC3B-positive structures (Fig. 3D).

These findings, together with the link between preferential AMPK and ERK1/2 activation downstream of CSF1R and the transcriptional profile, strongly suggest elevated autophagy activity of aged microglia. To evaluate this, we targeted *Ulk1*, a gene which encodes a protein required for induction of canonical autophagy, in mice expressing Cre recombinase under the *Cx3cr1* promoter to create an inducible microglia *Ulk1* deletion upon tamoxifen treatment (the strain *Ulk1*[fl/fl]CX3CR1[CreERT2] hereafter referred to as *Ulk1*[fl/fl])[76]. Approximately 20 months after *Ulk1* deletion, we observed an almost complete loss of the P-ERK1/2[High] microglia population

(Fig. 3E, F). This confirmed that the upregulated autophagy led to the altered phosphorylation pattern in the subpopulation of aged microglia, as originally hypothesized. The P-ERK1/2[High] population also had a reduced Akt phosphorylation (Fig. 3G) and proliferated less (Fig. 3H) than P-ERK1/2[Low] microglia. In vitro, *Ulk1* deletion in microglia had no effect on ERK1/2, AMPK and Akt phosphorylation, nor did this impact protein levels of ERK1/2, AMPK and Akt (Fig. S1D), suggesting, as expected, that ULK1 is a downstream target rather than a regulator of the pathway. *Ulk1*[fl/fl] microglia had reduced ULK1 protein levels in both adult as well as aged mice, validating the gene deletion and supporting the notion that an autophagy competent myeloid population does not replace the *Ulk1*[fl/fl] microglia population (Fig. S1C).

We hereby define a microglia subpopulation with activated ERK1/2 that appears with age and is dependent on canonical autophagy for survival. For simplicity, we henceforth refer to this population as "ADAM"- <u>A</u>utophagy <u>D</u>ependent <u>A</u>ge-acquired <u>M</u>icroglia.

### The loss of the reactive ADAM subpopulation is not compensated by other microglia or infiltrating myeloid cells

In order to investigate events underlying reduced ADAM population following impaired canonical autophagy, we analyzed microglial viability and mitosis ex vivo. We found the P-ERK1/2[High] microglia population in the aged *Ulk1*[fl/fl] strain to have an increased proportion of cells undergoing apoptosis (detected as AnnexinV[+]), while proliferation measured by EdU uptake was not affected (Fig. 4A, B). ER stress and CHOP associate through the unfolded protein response to increased apoptosis[47]. Thus, the ULK1 deletion effect on cell survival was further investigated in the context of ER stress and we found increased nuclear CHOP density in the aged *Ulk1*[fl/fl] microglia (Fig. 4C, D). Of note, the loss of ADAM was neither compensated by other microglia, detected as CD45[Intermediate] cells or by infiltration of peripheral CD45[High] myeloid cells (Fig. 4E, F)[77].

Immunohistochemical analysis revealed a notable decrease of the ADAM population, predominantly in the grey matter regions of the cerebral cortex (Fig. 4G, H, S2A, B). Within these regions, we observed an increased density of LC3 in the P-ERK1/2[High] population in Wt mice, which may suggest the initiation of autophagosome formation (Fig. 4I). We did not detect a significant reduction of ADAM or the microglia population at large in cerebral periventricular white matter regions or in the spinal cord, here analyzed at thoracic level (Fig. 4G, H, S2C). These findings align with earlier studies that associate IL-34 with microglia in grey matter and CSF-1 with those in white matter[19,20]. Notably, since the microglial population count was not affected in aged mice when Tamoxifen-induced deletion of *Ulk1* was done one month before they were sacrificed (Fig. 4E, F), we concluded that the reduction of the microglia, particularly ADAM population, was caused by a long-term cumulative loss. Concurrent with the loss of microglia, the expression of *Csf1* and *Il34* in the CNS was decreased (Fig. 4J)[16,17].

In summary, our findings are consistent with an impaired survival rather than insufficient mitosis explaining the reduced count of microglia numbers in the aged CNS of *Ulk1* deficient mice. The loss of

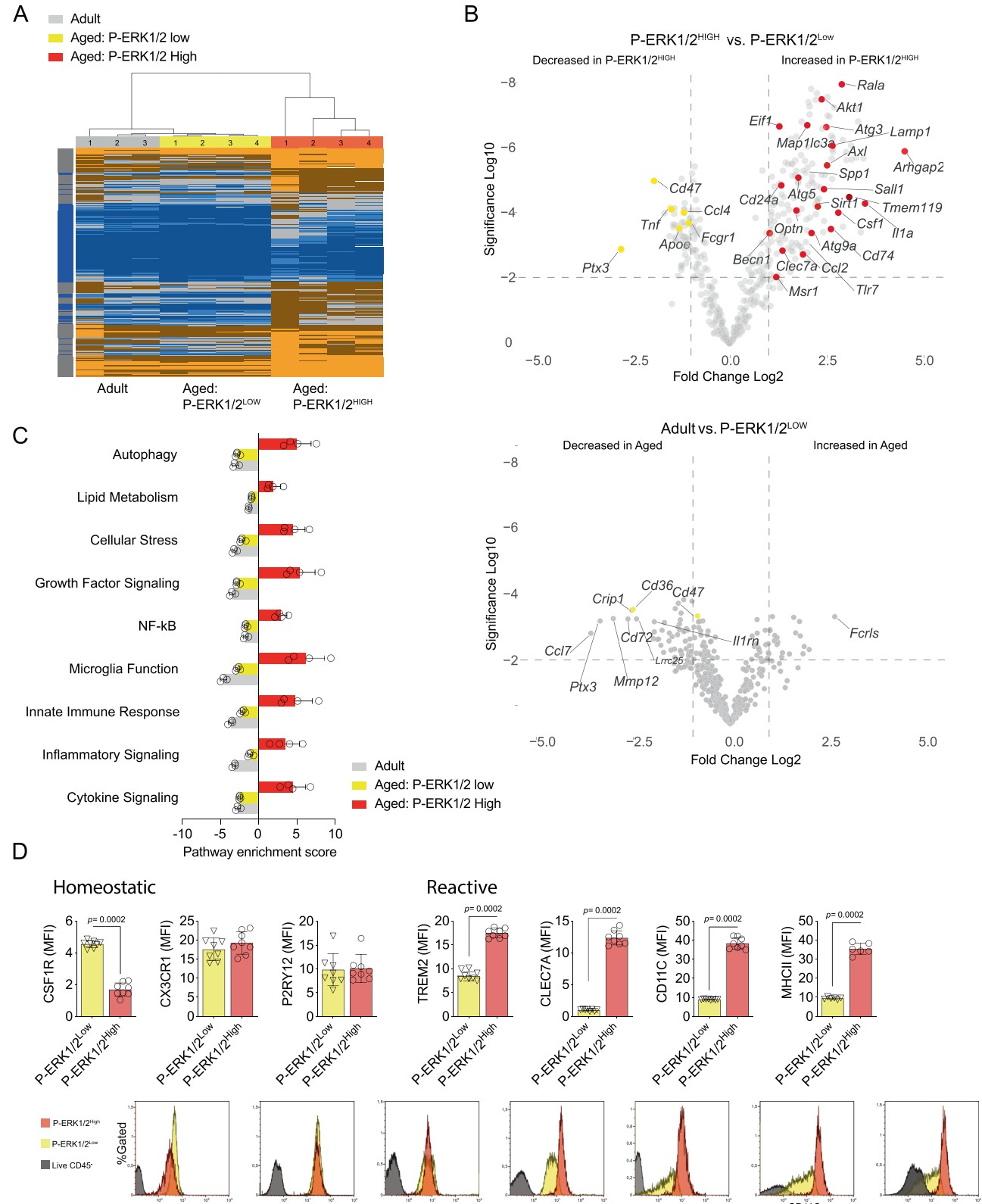

**Fig. 2 | The microglia of the aged CNS with reactive ERK1/2 pathway have a distinct transcriptional profile of microglial functionality. A** Transcriptome heatmap of normalized expression of all genes (770) analyzed by Nanostring™ nCounter (adult $n = 4$; aged P-ERK1/2$^{Low}$, $n = 4$; aged P-ERK1/2$^{High}$, $n = 4$). **B** Volcano plot of differentially expressed genes (Log2 > 1, p < Log10-2) comparing aged P-ERK1/2$^{Low}$ to P-ERK1/2$^{High}$ (upper) and aged ERK1/2$^{Low}$ to unsorted adult (lower) microglia. Populations described in **A**. **C** nSolver™ pathway transcriptome analysis of microglia populations described in **A**. **D** Microglial surface receptor density quantified ex vivo by flow cytometry in aged Wt subpopulations based on P-ERK1/2 (Thr202/Tyr204). (All $n = 8$). Adult mice were 3-5-month-old, <1 month post-Tamoxifen treatment, Aged mice were >20 months old, >18 months post Tamoxifen treatment. **D** Mann-Whitney Two-tailed $U$-test. Error bars represent mean + SD. Source data are provided as a Source Data file.

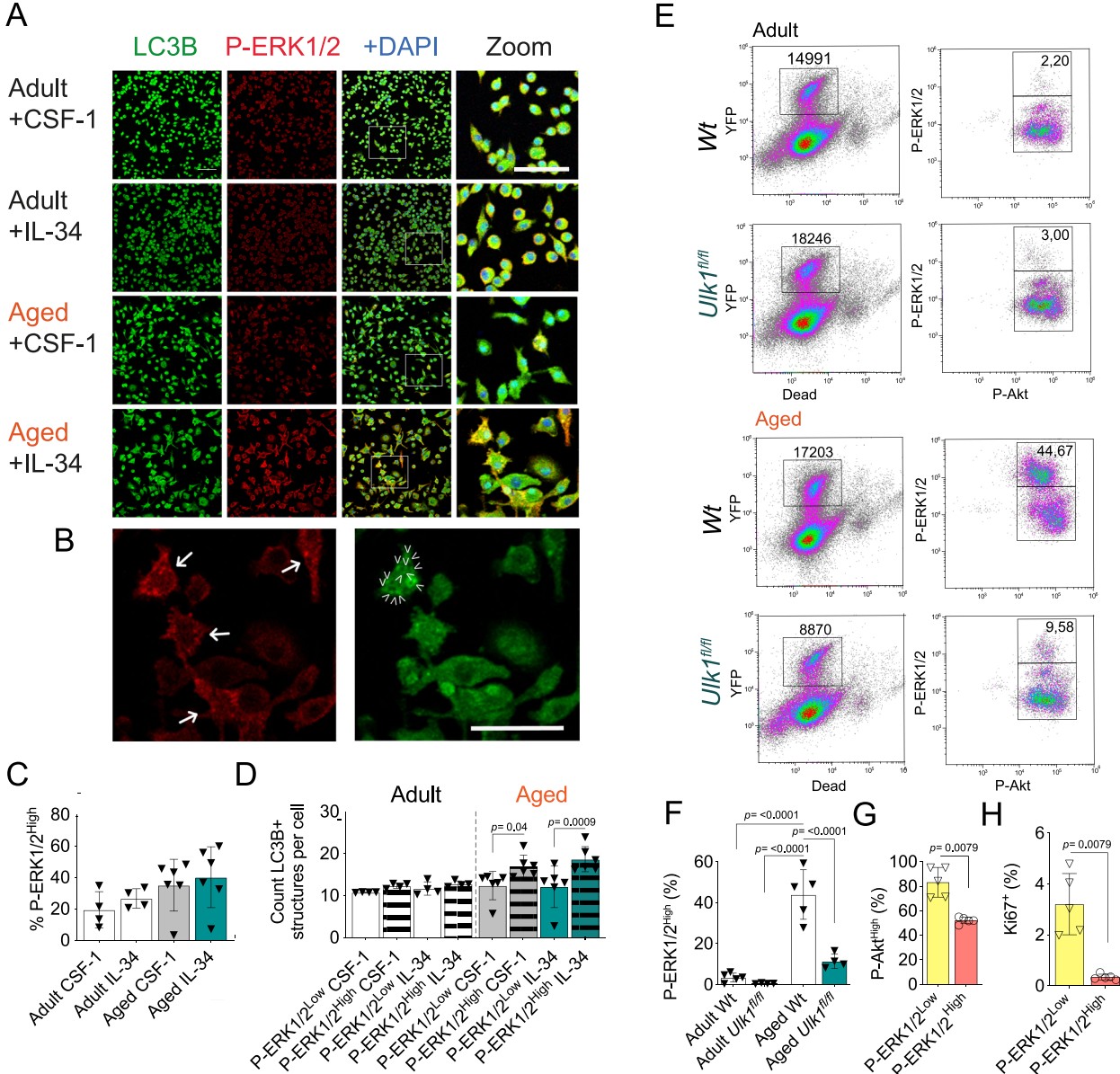

**Fig. 3 | Deficiency in canonical autophagy protein ULK1 leads to loss of an age-acquired microglial subpopulation.** **A, B** Immunocytochemistry images and quantification of microglial P-ERK1/2 high cells (Adult CSF-1 treated, $n = 4$; Adult IL-34 treated, $n = 4$; Aged CSF-1 treated, $n = 6$; Aged IL-34 treated, $n = 6$). Zoomed image with individual channels are found in supplementary fig. 2. Scale bars correspond to 50 µm. **C** Zoomed images shown in **A** with arrows indicating P-ERK1/2$^{High}$ cells (left) and one cell with arrows indicating structures detected as LC3B+ autophagosomes (right). **D** Quantification of LC3B+ autophagosomal structures in microglia defined by the P-ERK1/2 level. Representative images shown in **A**. **E** Microglial counts and AKT and ERK1/2 phosphorylation state at targeted sites assessed by flow cytometry. **F** Quantification of microglia populations by flow cytometry as shown in **E** (Adult Wt, $n = 5$; Adult $Ulk1^{fl/fl}$, $n = 4$; Aged Wt, $n = 5$; Aged $Ulk1^{fl/fl}$, $n = 4$) based on P-ERK1/2 (Thr202/Tyr204). **G** Akt (Ser473) phosphorylation in the P-ERK1/2-positive populations (Both groups $n = 4$) quantified by flow cytometry. **H** Microglia proliferation assessed by ex vivo Ki67+ staining and flow cytometry (Both groups $n = 5$). Adult mice were 3-5-month-old, <1 month post-Tamoxifen treatment, Aged mice were >20 months old, >18 months post Tamoxifen treatment. $Ulk1^{fl/fl}$ refer to $Ulk1^{fl/fl}$ CX3CR1$^{CreERT2}$ and Wt to $Ulk1^{wt/wt}$ CX3CR1$^{CreERT2}$. **A, B** and **G** showing representative data of two independent experiments. **C−F** showing representative data of three independent experiments. **B, C** and **E** One-way Anova followed by Tukey post-hoc test. **F−G** Mann-Whitney Two-tailed *U*-test. Error bars represent mean + SD. Source data are provided as a Source Data file.

the ADAM population specifically impacted microglia in the cortical brain regions, and this loss was not mitigated by the expansion of either the remaining microglial cells or infiltrating myeloid populations, while this autophagy-dependent population was found to be less proliferative but with a reactive phenotype.

## IL-34, but not CSF-1, enhances the ADAM population
The two known ligands of CSF1R have disparate spatial expression and alternate impact on microglial phenotypes[19,20,22,23]. By injecting these cytokines intrathecally to aged wild-type (Wt) and $Ulk1^{fl/fl}$ mice we

found both cytokines to expand the microglia population (Fig. 5A). This expansion was not dependent on ULK1 in the CSF-1 treated CNS, while IL-34 specifically expanded the ADAM population with increased P-ERK1/2 (Fig. 5A, B). Importantly, the reduction of the ADAM population did not induce peripheral BMDM immune cell infiltration in the untreated mice, neither did the CSF-1 or IL-34 administration (Figs. 4F and 5C). IL-34 can also bind Syndecan-1 and PTPRZ1 receptors[78]. Due to this, we wanted to study if the observed expansion of the P-ERK1/2$^{High}$ population was mediated through the CSF1R signaling. To this end, we treated ex vivo Wt microglia with CSF-1 and IL-34 in the presence of the

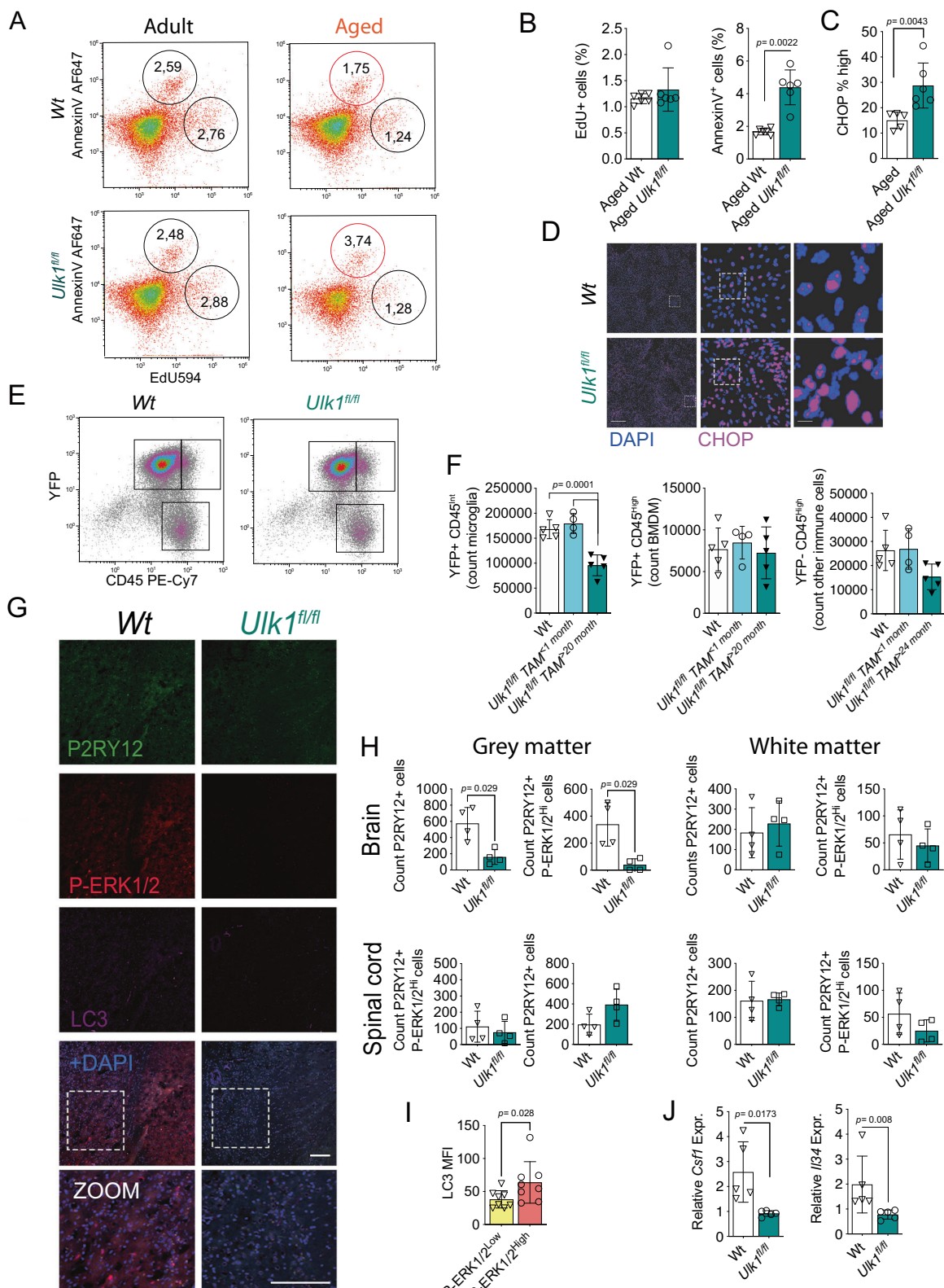

CSF1R inhibitor PLX3397. Upon this inhibition, we observed a complete loss of P-ERK1/2 (Fig. 5D). Thus, while IL-34, but not CSF-1, readily sustained the ERK1/2-Thr202/Thyr204 phosphorylation ex vivo, CSF1R antagonism abolished it (Fig. 5D)[16,17]. Neither of these treatments affected the levels of targeted protein (Fig. 5E).

In line with the expansion of the ADAM microglia, we could further show a maintained reactive microglia differentiation upon IL-34,

but not CSF-1, treatment in vivo (Fig. 5F). We also found a cytokine-dependent P2RY12 expression, although the elevation associated with the IL-34 treatment remained below that observed for the PBS control (Fig. 5F).

Previous studies reported no impact on neuronal and glial cell density after the treatment with IL-34 and CSF-1 neutralizing antibodies[79]. In line with that, we demonstrated that in naïve state

**Fig. 4 | The loss of the reactive ADAM subpopulation is not compensated by other cells. A, B** Microglia proliferation and apoptosis assessed by flow cytometry after three EdU⁺ i.p. injections in vivo days 5, 3 and 1 before analysis (adult, n = 6; aged, n = 6). **C, D** Immunocytochemistry images and quantification of CHOP positive ex vivo microglia cultured for 16 h. DAPI defines nuclei (Wt, n = 5; Ulk1ᶠˡ/ᶠˡ, n = 5). Scale bars correspond to 1000 μm (left) and 25 μm (right). **E, F** Flow cytometry density plots and quantification of CNS CD45⁺ populations (Wt, n = 5; Ulk1ᶠˡ/ᶠˡ, TAM < 1 month n = 4; Ulk1ᶠˡ/ᶠˡ TAM > 20 months n = 5) where CD45^Intermediate YFP⁺ represent microglia targeted by the Tamoxifen induced Ulk1 deletion. The YFP+ CD45^High populations are considered bone marrow-derived myeloid cells and YFP- CD45+ other immune cells. **G, H** Immunohistochemistry images and quantification of microglia in white and gray matter (shown in images) in brain and spinal cord regions (Brain, Wt, n = 5; Ulk1ᶠˡ/ᶠˡ, n = 4, spinal cord, Wt, n = 4; Ulk1ᶠˡ/ᶠˡ, n = 4). Scale bars correspond to 100 μm. Zoomed image with individual channels and images of spinal cords and white matter regions are found in supplementary fig. 2. **I** Quantification of MFI in microglia subpopulations from grey matter Wt images shown in **H**. **J** CNS mRNA expression of Csf1 and Il34 normalized to Hprt (CSF-1 Wt and Ulk1ᶠˡ/ᶠˡ, n = 5; Il34 Wt, n = 7; Ulk1ᶠˡ/ᶠˡ, n = 4). All experimental subjects **A–H** were aged (>20 months old, >18 months post-Tamoxifen treatment, except **C** > 20 months, <1 month Tamoxifen treatment). Ulk1ᶠˡ/ᶠˡ refer to Ulk1ᶠˡ/ᶠˡ CX3CR1^CreERT2 and Wt to Ulk1^wt/wt CX3CR1^CreERT2. **A, B** and **E, F** showing representative data of three independent experiments. **C** and **B** show data of two independent experiments **B, E** and **H–J**. **F** One-way Anova followed by Bonferroni's multiple comparisons test. Mann-Whitney Two-tailed U-test. (**H**). Error bars represent mean + SD. Source data are provided as a Source Data file.

neither of these treatments or the loss of ADAM affected the counts of CD171⁺ neurons, MBP⁺ oligodendrocytes (OLs) or PDGFRa⁺ oligodendrocyte progenitor cells (OPCs) (Fig. 5G)[19].

## Loss of ADAM associates with higher mortality during autoimmune neuroinflammation

Microglia at large, and reactive subpopulations such as DAM, have been assigned both detrimental and protective phenotypes during neuroinflammation[80]. More precise conditions for these variable effects of microglia need to be defined and are called for by emerging concepts of myeloid modulatory therapies. We sought to challenge the CNS with myelin oligodendrocyte glycoprotein (MOG)-induced EAE, a model widely used to understand underlying mechanisms and develop therapies for MS[81], to evaluate the impact of the reduced microglial population and the specific loss of ADAM. The increased mortality of Ulk1ᶠˡ/ᶠˡ aged mice compared to Wt mice was remarkable, indicating a protective role of ADAM (Fig. 6A). The CSF-1 treatment had no significant effect on the disease course of Ulk1 deficient mice, even though the total microglia population expanded to Wt levels (Fig. 5A). Moreover, the Ulk1ᶠˡ/ᶠˡ mice displayed reduced microglial numbers despite increased proliferation, which was counteracted by high apoptosis (Fig. 6B). During neuroinflammation, the Ulk1ᶠˡ/ᶠˡ mice showed reduced numbers of neurons, OLs and OPCs in the CNS, and the main target for the auto-aggression, i.e. the MBP⁺ OLs showed a significantly increased proportion of apoptotic cells (Fig. 6B, Fig. S3A).

Immunohistochemical analysis unveiled a pronounced augmentation in the demyelinated lesion load in Ulk1ᶠˡ/ᶠˡ mice lacking the ADAM population (Fig. 6C, D). These demyelinated regions exhibited a heightened prevalence of CD68⁺ macrophages, presumably originating from monocytes (Fig. 6C). Additionally, conspicuous deposits of amyloid precursor protein (APP) were evident within the tissue, indicative of neuronal pathology[82] (Fig. 6C). Notably, these findings manifested primarily within the cerebral tissues of Ulk1ᶠˡ/ᶠˡ mice, while the spinal cord exhibited an absence of pathological alterations. This spatial discrepancy is congruent with the divergent influence exerted by IL-34 versus CSF-1, and aligns with the lethal EAE phenotype detected. CNS of the Ulk1ᶠˡ/ᶠˡ mice also exhibited extensive macrophage activation in tissue outside demyelinating lesions, suggesting a repopulation of previously vacant niches (Fig. S3B). Throughout the course of EAE, we observed a significant increase in the expression of Csf1 but not Il34. This finding underscores the widely recognized inflammatory responsiveness associated with Csf1 (Fig. S3C).

In conclusion, naïve ADAM-deficient mice did not show reduced counts of neurons or cells of the oligodendrocyte lineage. However, when challenged with EAE the remaining microglia failed to protect the CNS cells from death and EAE-associated mortality.

## IL-34 administration expands ADAM, restricts autoimmune neuroinflammation and reduces its clinical signs

While intrathecal injections of IL-34 and CSF-1 did not rescue the Ulk1ᶠˡ/ᶠˡ mice during EAE, aged Wt mice displayed a significant amelioration of disease upon IL-34 treatment (Fig. 7A). This was associated with reduced infiltration of peripheral BMDMs and T cells accompanied by an increased microglial counts (Fig. 7C). The IL-34 treatment also rescued neurons and OLs in the CNS during EAE compared to PBS and CSF-1 treatments, a trend also seen for OPCs (Fig. 7E–G). The beneficial effects of IL-34 treatment were all abrogated in Ulk1ᶠˡ/ᶠˡ mice, which are lacking the age-acquired ADAM population (Fig. 7A–E). Like untreated EAE mice, Ulk1ᶠˡ/ᶠˡ mice showed increased immune cell infiltration and reduced neuron, OL and OPC counts with a significant proportion of apoptotic cells in all targeted CNS resident cells (Fig. 7C, E). Microglial phagocytosis plays a pivotal role in mitigating CNS damage across various pathological conditions, including EAE[83–85]. In our investigation, we observed an elevated surface density and increased expression of scavenger receptors within the ADAM population, as illustrated in Fig. 2. Hence, we aimed to ascertain whether this phagocytic capability underwent alterations in the ADAM population. To do so, we conducted an ex vivo experiment, exposing microglia from aged mice to fluorophore-conjugated myelin debris. Surprisingly, our analysis revealed a non-significant difference in the uptake of myelin debris when comparing the P-ERK1/2^High and P-ERK1/2^Low subsets although the autophagosome marker LC3 was detected at higher density in ADAM (Fig. 8A). This observation suggests the existence of additional neuroprotective functions within the ADAM population. To address the mechanism behind the neuroprotection following IL-34 treatment, we co-cultured neurons with microglia sorted from aged Wt and Ulk1ᶠˡ/ᶠˡ mice and exposed them to CSF-1 or IL-34 or unstimulated control. IL-34 treatment increased neuron counts and reduced the number of apoptotic neurons when co-cultured with Wt microglia but not Ulk1-deficient microglia (Fig. 8B–D). We also detected increased counts of microglia in the IL-34 treated condition (Fig. 8B). Of note, CSF-1 or IL-34 did not affect neuronal apoptosis or counts in cultures without microglia (Fig. 8D). Finally, we found microglia from IL-34 treated co-cultures to express higher mRNA levels of neurotrophic factors, such as Bdnf, Igf1, Tgfb1 and Manf compared to only medium and cultures with Ulk1 deficient microglia. CSF-1 had an intermediate effect on the above-mentioned targets as well and a overall tendency for lower expression compared to IL-34 treated microglia. These represent microglia-derived neuroprotective mediators, possibly explaining the health-promoting microglial phenotype associated with ameliorated neuroinflammation where IGF1 and BDNF been associated to MS risk and the latter also linked to microglial autophagy[60,86–89].

Taken together, these data strongly suggest a neuroprotective ADAM-dependent effect of IL-34 in ameliorating EAE and neuronal survival in vivo and in vitro.

## Discussion

We here identify and define an autophagy-dependent microglial population in the aging CNS with health-promoting capacities required for survival from neuroinflammatory and neurodegenerative disease. Phenotyping of this cortical microglia subpopulation revealed a neuroprotective function and a pattern of activation seen in other models

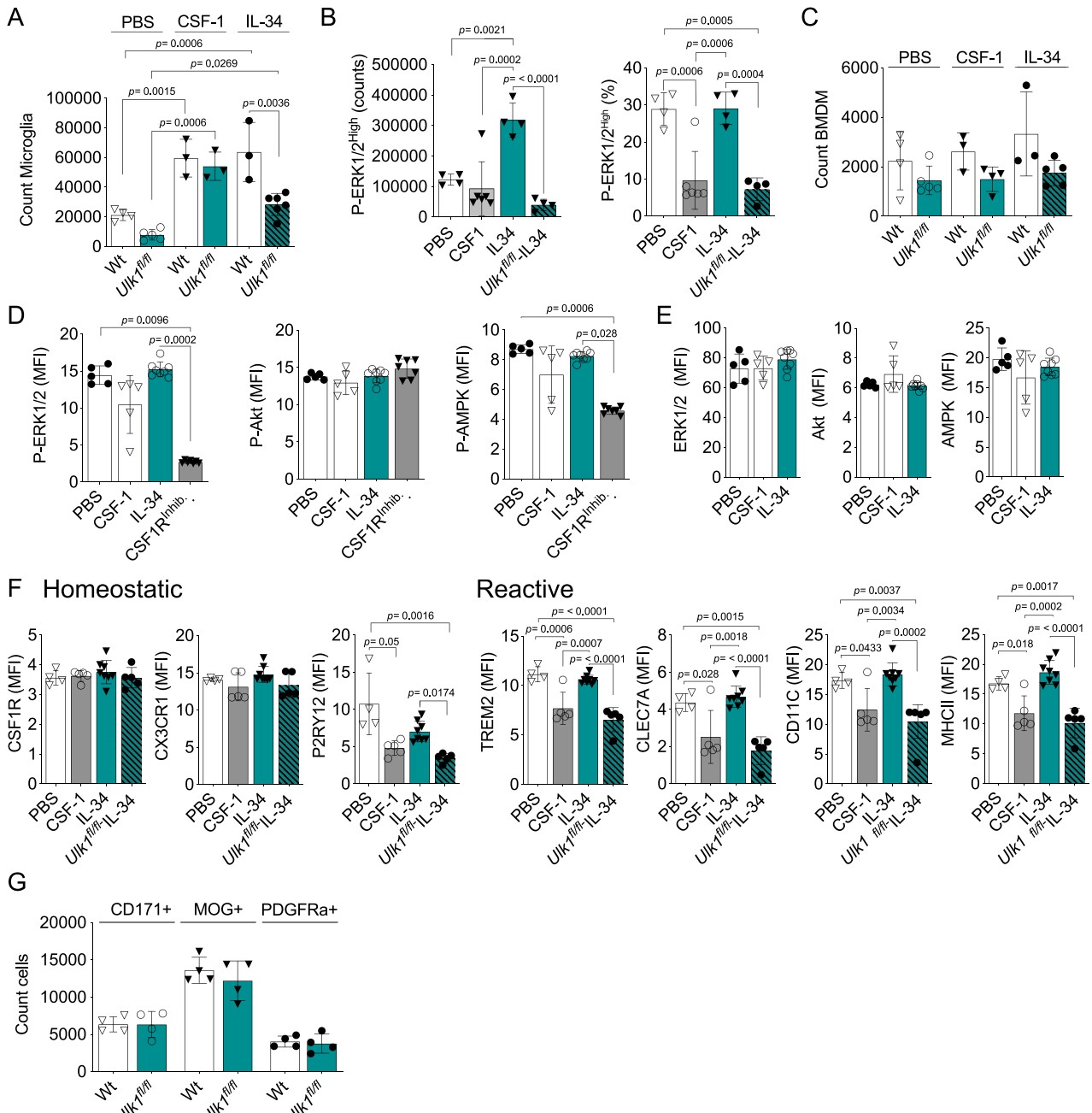

**Fig. 5 | IL-34 but not CSF-1 enhance the ADAM population. A** Microglia (CD11b+ CD45Int YFP+) counts from mice treated intrathecally with IL-34, CSF-1 or PBS control (PBS Wt, *n* = 4; PBS *Ulk1fl/fl*, *n* = 5; CSF1 Wt, *n* = 3; CSF1 *Ulk1fl/fl*, *n* = 3; IL-34 Wt, *n* = 3; IL-34 *Ulk1fl/fl*, *n* = 6). **B** P-EKR1/2-positive microglia cells as percent of the total microglia population from mice treated as in (A) (PBS Wt, *n* = 4; CSF-1 Wt, *n* = 6; IL-34 Wt, *n* = 4; IL-34 *Ulk1fl/fl*, *n* = 4). **C** BMDM (CD11b+ CD45High) counts from mice treated as in **A**. **D** Phosphorylation and total content of target proteins in microglia treated ex vivo with CSF1R ligands individually or with both ligands in the presence of the CSF1R inhibitor PLX3397 (PBS Wt, *n* = 5; CSF-1 Wt, *n* = 5; IL-34 Wt, *n* = 7; CSF1R^Inhib., *n* = 5). **E** Total content of target proteins in microglia treated ex vivo with CSF1R ligands individually (PBS Wt, *n* = 5; CSF-1 Wt, *n* = 5; IL-34 Wt, *n* = 7). **F** Homeostatic and reactive microglia surface markers assessed by flow cytometry

in freshly isolated microglia from mice treated as in **A** (PBS, *n* = 4; CSF-1, *n* = 5; IL-34, *n* = 7; *Ulk1fl/f*- IL-34, *n* = 5). **G** CD171+ neurons, MBP+ Oligodendrocytes and PDGFRa+ OPC counts (*n* = 4 in all conditions). Analyses separated per cell type. All experimental subjects **A**–**H** were aged (>20 months old, >18 months post-Tamoxifen treatment), all treatments refer to intrathecal administration 5, 3, and 1 day before analysis. *Ulk1fl/fl* refer to *Ulk1fl/fl* CX3CR1^CreERT2 *and Wt* to *Ulk1wt/wt* CX3CR1^CreERT2.
**A**–**D** and **F** showing representative data of three independent experiments. **E** and **G** showing representative data of two independent experiments. **D** and **E** Kruskal-Wallis followed by Dunn's multiple comparisons test (**A**) Two-way Anova followed by Šídák's multiple comparisons test (**B**) and (**F**) One-way Anova followed by Tukey or Bonferroni post-hoc test. **G** Mann-Whitney Two-tailed *U*-test. Error bars represent mean + SD. Source data are provided as a Source Data file.

of CNS pathology[54,57,58]. Intrathecal administration of the CSF1R ligand IL-34 expanded this population with an ameliorating impact on disease pathology and clinical manifestations.

We demonstrated that aged compared to adult microglia have a slower turnover rate and phenotypic alterations similar to microglia/macrophage signatures and functions previously described in

demyelinating and degenerative CNS diseases[9,56–58,68,77,85,90–93]. In advanced age microglia, we detected an altered downstream phosphorylation pattern of ERK1/2, Akt and AMPK, which signal downstream CSF1R that is fundamental for microglial viability. This changed phosphorylation pattern was associated with increased canonical autophagy. Several lines of evidence, including the known regulatory

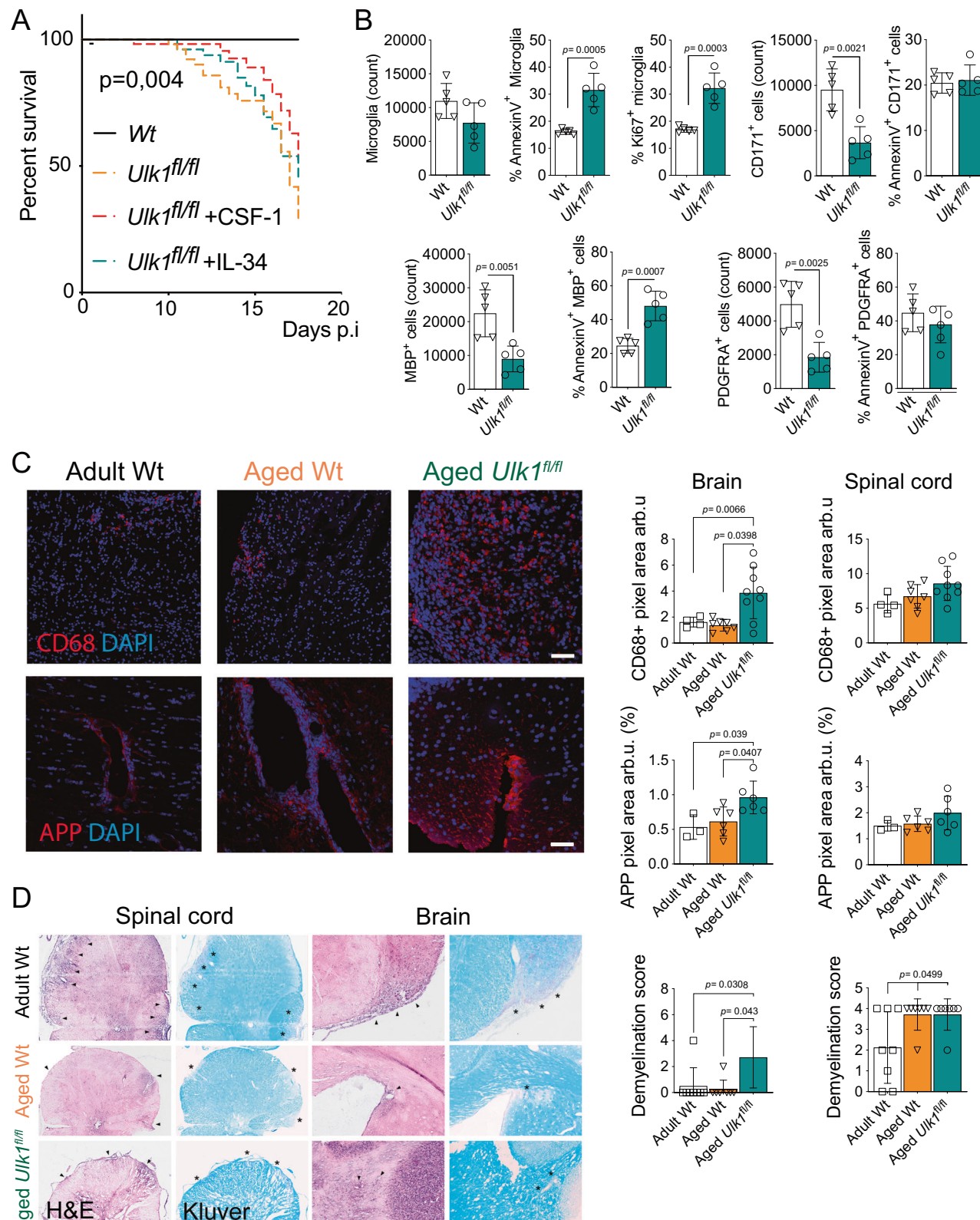

functions of detected phosphorylation patterns together with validation of autophagy activation by transcriptional analysis and ex vivo correlation between elevated P-ERK1/2 levels and increased LC3B⁺ compartment density, suggest enhanced autophagy activation, although we cannot exclude additional influence of reduced lysosomal degradation.

Accordingly, deletion of autophagy core gene *Ulk1* caused a loss of this age-acquired microglia population (referred to as ADAM) and a consequential reduction of the total microglia in the CNS of 2-year-old mice. The *Ulk1* deletion specifically impaired the survival of this population which associated with an increase in ER stress, consistent with the essential function of canonical autophagy[28,94,95]. The findings

**Fig. 6 | Loss of ADAM associates with higher mortality during autoimmune neuroinflammation. A** Survival plots after MOG-induced EAE. **B** Quantification of microglia numbers, apoptosis and proliferation and CD171⁺ neurons, MBP⁺ Oligodendrocytes and PDGFRA⁺ OPCs numbers and apoptosis at day 7 post-EAE onset (Wt, *n* = 5; *Ulk1*^fl/fl^, *n* = 5). **C** Immunohistochemical images (images showing brain lesion areas, unaffected parenchyma is shown in Fig S3B) and analysis of brain and spinal cord sampled day 15 post EAE induction (Wt adult, *n* = 3 for APP and *n* = 4 for CD68, Wt aged, *n* = 6; *Ulk1*^fl/fl^ *aged*, *n* = 8). Scale bars represents 125 μm for APP and 250 μm for CD68 and imaged are 2D renderings of thick Z-stacks.
**D** Immunohistochemical images and analysis of brain and spinal cord sampled day

15 post EAE induction. Black arrowheads indicate inflammatory infiltrates (Wt adult, *n* = 7, Wt aged, *n* = 6; *Ulk1*^fl/fl^ *aged*, *n* = 6). Scale bars represent 100 μm in **C** and 200 μm in **D**. Adult mice were 3-5-month-old, <1 month post-Tamoxifen treatment, Aged mice were > 20 months old, >18 months post Tamoxifen treatment. *Ulk1*^fl/fl^ refer to *Ulk1*^fl/fl^ CX3CR1^CreERT2^ *and Wt* to *Ulk1*^wt/wt^ CX3CR1^CreERT2^. **A** and **B** showing representative data of three independent experiments. **A** Mantel-cox logrank test of survival curves. **B, D** Mann-Whitney Two-tailed *U*-test (**C, D**) One-way Anova with Tukey posttest. Error bars represent mean + SD. Source data are provided as a Source Data file.

---

of a diminished microglia population challenge the common idea that the myeloid CNS niche must be replenished under all conditions by either microglia or bone marrow-derived macrophages[12,96–98]. However, the foundation of this idea relies on models associated with acute extensive apoptosis, which induces inflammation associated to immune cell proliferation[98,99]. In our model, the cumulative reduction of microglia is seen only at an advanced age and could arguably be considered immunologically more silent, similar to the human pathologies seen in patients with CSF1R mutations[73]. This conclusion is supported by recent findings showing substantial reduction of the microglia population upon deficiency of autophagy gene *Atg4b*[84].

The ADAM population was characterized by increased density of receptors associated with microglial phenotypes found in models of neurodegeneration and regarded as cells in a reactive state. Additional to autophagy, transcriptome analysis highlighted microglial functions such as phagocytosis as well as hallmark microglia markers *Tmem119 and Sall1*. This phenotype partially overlaps with microglial signatures described in neuroinflammation and neurodegenerative diseases and includes genes such as *Itgax, Clec7a* and genes associated with APC functions, commonly thought of as inflammatory[56–58,90,91]. However, the loss of this reactive and supposedly inflammatory microglia was detrimental leading to high mortality during autoimmune neuroinflammation. The underlying pathology was characterized by an increased peripheral immune cell infiltration accompanied by a neural and glial cell loss and demyelination during EAE. This supports the notion that the premise of a pro-inflammatory microglial phenotype as harmful needs to be revisited and contextualized. The signature of the ADAM population does not completely match other previously defined microglia populations and states. This could arguably be secondary to the P-ERK1/2 discrimination used in our study, but also supportive of microglia as dynamic cells shaped by demands of the aged brain.

Impaired autophagy has been associated with age or aging cells, but with the recent expansion of the concept and multitude of functions of autophagy proteins, we need to be open to the idea that the autophagy proteins and functions are employed in a cell- and context-dependent manner, rather than simply being up- or down-regulated[40,42,75,100]. We previously reported that non-canonical autophagy in microglia, which decreases with age, is essential for phagocytosis and recovery from MS-like neuroinflammation[75]. Of note, in the previous publication, the aged mice had *Atg7* or *Ulk1* genes ablated just before the induction of EAE, while here the mice were left to age without *Ulk1* in their microglia, which leads to a loss of the protective ADAM population.

Here we further characterized the microglial niche and demonstrated that aging leads to the expansion of a microglia population that relies on canonical autophagy for survival and neuroprotective functions in the face of inflammatory insults. This raises the question of competitive recruitment to these disparate processes underlying the impaired non-canonical autophagy in the aged microglia population. On the other hand, the autophagy-associated phagocytic capacity of the microglia population during recovery can represent a temporary shift in the autophagy protein employment in the ADAM population. A long-term commitment of the autophagy-associated proteins to one of

these functions could, however, potentially affect the microglia population's phenotype, function and maintenance.

Our findings present a nuanced perspective on the prevailing notion that age inherently constrains autophagy function, particularly within microglia. While this assertion currently lacks robust supporting evidence, it is plausible that autophagy function varies within the microglial phenotype spectrum. Overall, there may be a reduced autophagic capacity, but our observations suggest the existence of two distinct subpopulations; one characterized by low P-ERK1/2 levels, likely comprising less differentiated cells with low autophagy activity, and another with high P-ERK1/2 levels, representing the subset of functional cells that persist.

Recent efforts have defined unique microglial subpopulations in homeostatic and disease conditions, often with age as an influential factor. In these studies, CSF1R is associated with homeostatic microglia and has two known ligands, CSF-1 and IL-34[20,23,58,77,79,93]. While CSF-1 exclusively engages the CSF1R receptor, IL-34 exhibits binding affinity towards two additional identified receptors: phosphatase-ζ (PTP-ζ) and syndecan-1. The implications of these interactions on microglial function, although not further addressed in this study, warrant additional investigations[101]. While CSF-1 associates with inflammation and is secreted mainly by microglia and other immune cells, IL-34 is sourced by neurons and glial cells and is more abundant in steady-state but is yet to be comprehensively explored in neuroinflammatory diseases. Studies disrupting IL-34 and CSF-1 gene expression or treatment with specific blocking antibodies suggest regional dependency for these cytokines, with IL-34 affecting mostly the grey matter[19,23]. Accordingly, the observed absence of microglia was primarily detected in the cortical areas of the brain, an observation that potentially could elucidate the underlying factors contributing to the observed mortality.

By intrathecal injections of CSF-1 or IL-34, we expanded the microglia population in aged wild-type mice. While IL-34 expanded the responsive ADAM population and maintained the highly phosphorylated ERK1/2 found in this phenotype, CSF-1 induced the highly proliferative/less reactive population. The exact mechanisms underlying this discrepancy are not addressed in this study but could be influenced by the higher IL-34 affinity to CSF1R, which affects auto-phosphorylation and downstream regulation[26]. Further, in the aged *Ulk1* deficient mice, IL-34 failed to expand the population establishing support for an IL-34-CSF1R-ERK/AMPK-ULK1 axis.

The polarizing potential of IL-34 versus CSF-1 in previous studies indicates minor phenotype deviations upon alternate CSF1R ligands[19,22,23,26,66,102]. In this study, we found dramatic differences in several markers of functional differentiation such as CLEC7A and *Spp1*. However, the previous studies were conducted on young adult mice in which we did not observe the ADAM population. Intrathecal pretreatment specifically with IL-34 ameliorated EAE in aged wild-type mice, with reduced immune cell infiltration and increased survival of neurons and oligodendrocyte lineage cells. This effect was absent in mice with reduced ADAM population upon *Ulk1* deficiency and not significant upon CSF-1 treatment.

The pathology observed during EAE predominantly affected the brain instead of the spinal cord, a phenotype rarely seen in EAE. This observation supports the concept of a protective microglial

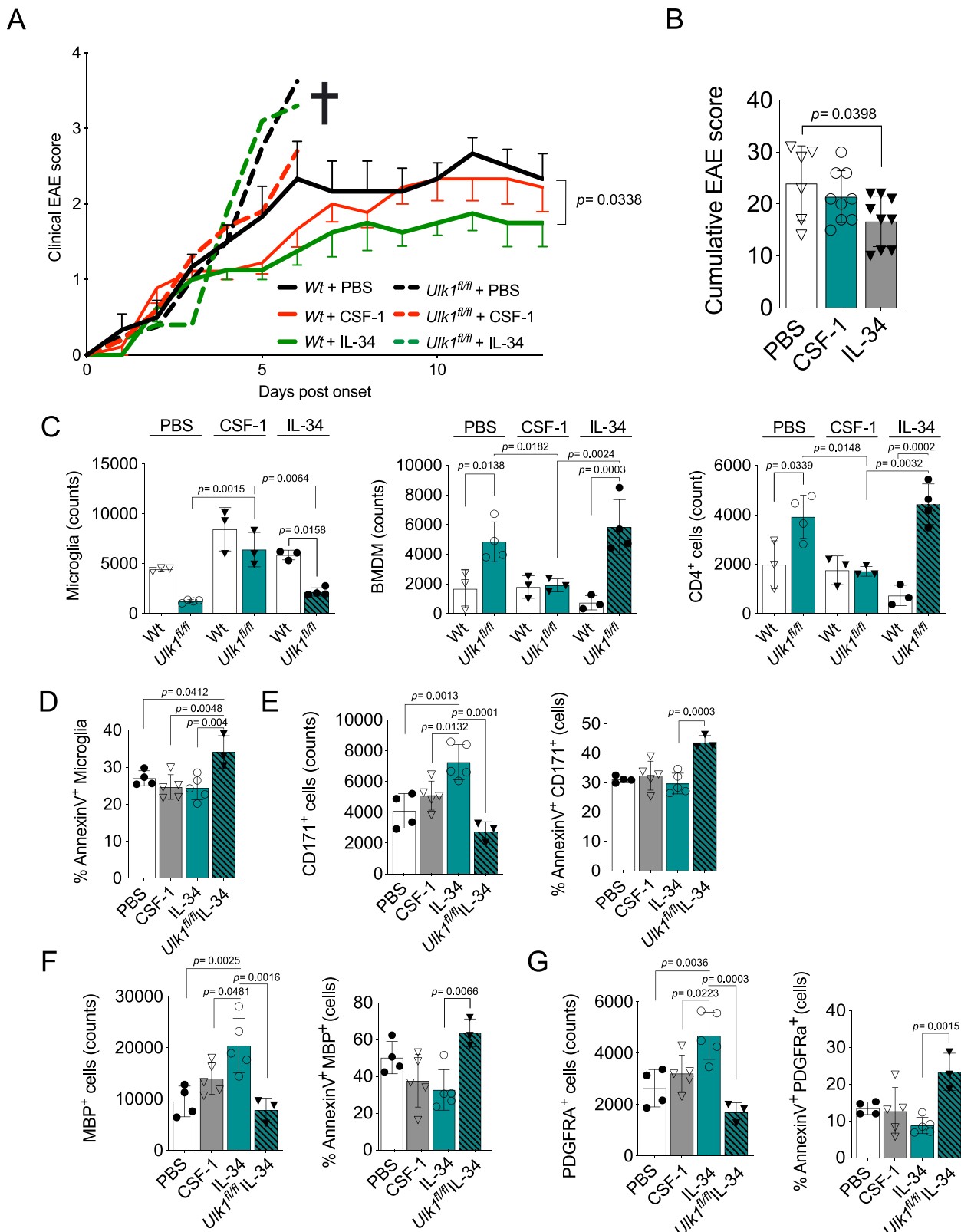

phenotype associated with IL-34, which is primarily expressed in cortical brain regions. Knowing that microglia are the main targets of IL-34 we believe the protective effect to be secondary to expansion of the ADAM population, a statement supported by the absent effect in mice with *Ulk1* deficiency. We could further validate a neuroprotective phenotype of the IL-34 treated microglia in vitro by cell viability measurements, possibly explained by increased expression of

neurotrophic factors. Notably we did not detect an increased increase in phagocytic capacity of the ADAM microglia, a function usually associated with DAM phenotypes and health promoting capacities of microglia in general. These experiments were however measuring specifically myelin uptake and lysosomal loading which do not exclude altered phagocytosis of e.g. apoptotic neurons. These findings are potentially highly relevant in human diseases where CSF-1 protein

**Fig. 7 | IL-34 administration expands ADAM, restricts autoimmune neuroinflammation and reduces its clinical signs. A** Clinical scores (left) and a graphic summary of aged mice with MOG-induced EAE who received intrathecal administration of CSF-1, IL-34 or PBS (Wt-PBS, $n = 6$; $Ulk1^{fl/fl}$-PBS $n = 6$; Wt-CSF-1, $n = 6$; $Ulk1^{fl/fl}$-CSF-1 $n = 10$; Wt-IL-34, $n = 6$; $Ulk1^{fl/fl}$-IL-34 $n = 8$. **B** Cumulative EAE scores of conditions described in (A) CSF-1, IL-34 or PBS (Wt-PBS, $n = 6$; Wt-CSF-1, $n = 9$; Wt-IL-34, $n = 9$. **C** Microglia, BMDM and CD4$^+$ T-cell counts day 7 post-EAE onset assessed by flow cytometry. (Wt-PBS, $n = 4$; Wt-CSF-1, $n = 5$; Wt-IL-34, $n = 5$; $Ulk1^{fl/fl}$-IL-34 $n = 3$). **D–G** Microglia, CD171$^+$ neurons, MBP$^+$ oligodendrocytes, PDGFRa$^+$ OPCs and their respective AnnexinV$^+$ subpopulations day 7 post-EAE onset assessed by flow cytometry. (Wt-PBS, $n = 3$; $Ulk1^{fl/fl}$-PBS $n = 4$; Wt-CSF-1, $n = 3$; $Ulk1^{fl/fl}$-CSF-1 $n = 3$; Wt-IL-34, $n = 3$; $Ulk1^{fl/fl}$-IL-34 $n = 4$. All experimental subjects (**A–E**) were aged (>20 months old, >18 months post-Tamoxifen treatment), all treatments refer to intrathecal administration 5, 3, and 1 day before disease induction. $Ulk1^{fl/fl}$ refer to $Ulk1^{fl/fl}$ CX3CR1$^{CreERT2}$ and Wt to $Ulk1^{wt/wt}$ CX3CR1$^{CreERT2}$. **A–G** showing representative data of three independent experiments. **A, B** and **D–G** One-way Anova followed by Bonferroni's, Tukeys or Dunn's multiple comparisons test. **C** Two-way Anova followed by Šídák's multiple comparisons test. Mann-Whitney Two-tailed $U$-test. Error bars represent mean + SD. Source data are provided as a Source Data file.

levels are increased while IL-34 concentration is not affected, such as in the cerebrospinal fluid of patients suffering from progressive MS[24]. This could favor a CSF-1-expanded quiescent over the IL-34-polarized neuroprotective microglia population, contributing to progression. Although sex is acknowledged to impact neuroinflammation and microglial phenotypes[103], we did not observe any sex-specific phenotypes in subgroup analysis of our key experiments (Supplementary fig. 4).

We support the idea of microglia as long-lived health-promoting cells with a dynamic nature as an attractive treatment target. Although the EAE model has limitations in translation to human MS disease, the mechanisms shown here could contribute to the age-related conversion to progressive disease that could be targeted towards addressing this major challenge in managing MS. Given the expanding idea of a therapeutic reconstitution of microglia in age-associated diseases using CSF1R inhibition, our findings propose a need to take IL-34 into account in shaping the new protective population state. The autophagy-dependency of the health-promoting microglia defined in these studies should also be viewed as a possible pathway to strengthen the CNS innate immunity during aging.

## Methods

### Ethics Statement

Animal experiments were approved and performed according to regulations from the Swedish National Board for Laboratory Animals and the European Community Council Directive (86/609/EEC) under the ethical permits N338/09, N138/14, and 9328-2019, approved by the North Stockholm Animal Ethics Committee (Stockholms Norra djurförsöketiska nämnd). Mice were regularly tested according to a health-monitoring program from the National Veterinary Institute (Statens Veterinärmedicinska Anstalt, SVA) in Uppsala, Sweden.

### Experimental Subjects

Gene-deleted mice on the C57BL/6 background were generated by breeding mice with a floxed $Ulk1$ gene ($Ulk1^{fl/fl}$) to mice with Cre recombinase expressed under the $Cx3cr1$ promoter ($Cx3cr1^{CreERT2}$). All strains were purchased from The Jackson Laboratory. All mice were housed at animal facilities at Karolinska Institutet and bred and maintained under specific-pathogen-free conditions in polystyrene cages with hiding places in accordance with national animal-care guidelines. Housing rooms had 12 hour light/dark cycle, temperature was 20-24 °C and humidity 45-65%. Mice had access to standard chow and drinking water ad libitum. During EAE mice had access to wet food. Aged mice refer 20-25 months old and Adult mice 3-5 to months old. The experimental Cre mice had a hemizygote $Cx3cr1^{CreERT}$ genotype. Tamoxifen (TAM; Sigma), 4 mg dissolved in corn oil was administered subcutaneously three times in 48-hour intervals in 1 to 3 month old mice except in Fig. 3 were mice older than 20 months had Tamoxifen treatment 1 month prior the experiment. Experiments targeting microglia were started earliest 4 weeks after the last Tamoxifen administration in order to allow repopulation of peripheral bone marrow-derived $Cx3cr1^{CreERT2}$ expressing cells, e.g., monocytes. At the same time, the gene deletion effect is preserved and carried on in the self-renewing microglial population. Other CNS-associated self-renewing macrophages residing in meninges and perivascular spaces that also express that $Cx3cr1$ have been shown not to contribute to T cell activation and CNS damage during EAE and are not found in the parenchyma[91,104]. $Cx3Cr1$ positive cells also express fluorescent eYFP. No noticeable Cre$^{+/-}$ toxicity was observed. We used 3-5 months old littermate "Adult" mice and > 20 months old "Aged" mice in experiments. The $Ulk1$ deletion in aged mice was initiated at least 18 months prior to the commencement of the experiments, with the exception of Fig. 4h, where we specifically investigated the consequences of both long-term and short-term $Ulk1$ deficiency. Injections of 50 ng CSF-1 or IL-34 in 10uL saline were made intrathecally (described in[105]) for three consecutive days before sacrifice or immunization for EAE. For all experiments both female and male mice were used in matched groups. For IL-34 and CSF-1 treatment experiments mice were randomized from all cages. No uneven distribution of weight at start of experiment was detected. For all experiments, mice were euthanized with $CO_2$. Both females and males were used in all experimental groups, gender is annotated in the Source data file as well as subgroup analysis based on sex for key experiments.

### Induction and clinical evaluation of EAE

Recombinant mouse myelin oligodendrocyte glycoprotein (rmMOG) aa1-125 from the N-terminus, was produced in *Escherichia coli* and purified using chelate chromatography, as previously described[106,107]. EAE was induced by immunizing mice under isoflurane (Baxter) anesthesia with a single subcutaneous injection at the dorsal tail base with 100 μl of inoculum containing rmMOG 20-35 μg/mouse in PBS emulsified in a 1:1 ratio with Complete Freund's Adjuvant (CFA, Chondrex) (100 μg Mycobacterium tuberculosis/mouse). Emulsions were prepared using POWER-Kits purchased from BTB Emulsions, Malmö, Sweden (https://btbemulsions.com/), according to the manufacturer's recommendations. Additionally, all experimental animals received an i.p. administration of 200 ng/mouse pertussis toxin (PTX, Calbiochem) at days 0 and 2 p.i. The clinical disease was scored as follows: 0, no clinical signs of EAE; 1 - tail weakness or tail paralysis; 2 - hind leg paraparesis or hemiparesis; 3 - hind leg paralysis or hemiparalysis; 4 - tetraplegia or moribund; 5 - death. Ethical endpoints (weight loss, score 4, signs of bladder infections or wounds) stated in permits were applied. No mice were excluded from analysis based on these or other parameters. Following the onset of the disease, mice were provided with wet food due to their compromised ability to forage for both food and water, due to motor impairments.

### Single cell suspensions from CNS

CNS cells were extracted using the Neural Tissue dissociation kits T or P (Miltenyi Biotec).

Mice were anesthetized with isoflurane and perfused transcardially with ice-cold PBS. Brain and spinal cord were mechanically dissociated and resuspended in enzyme mix according to the manufacturer's protocol. The pellet was resuspended in a 38% Percoll (Sigma) solution and centrifuged at 800 g for 15 min (no brake). The myelin gradient layer was discarded, and the cell pellet resuspended in PBS. Alternatively, Myelin removal beads were used according to instructions from the manufacture (Miltenyi Biotec.)

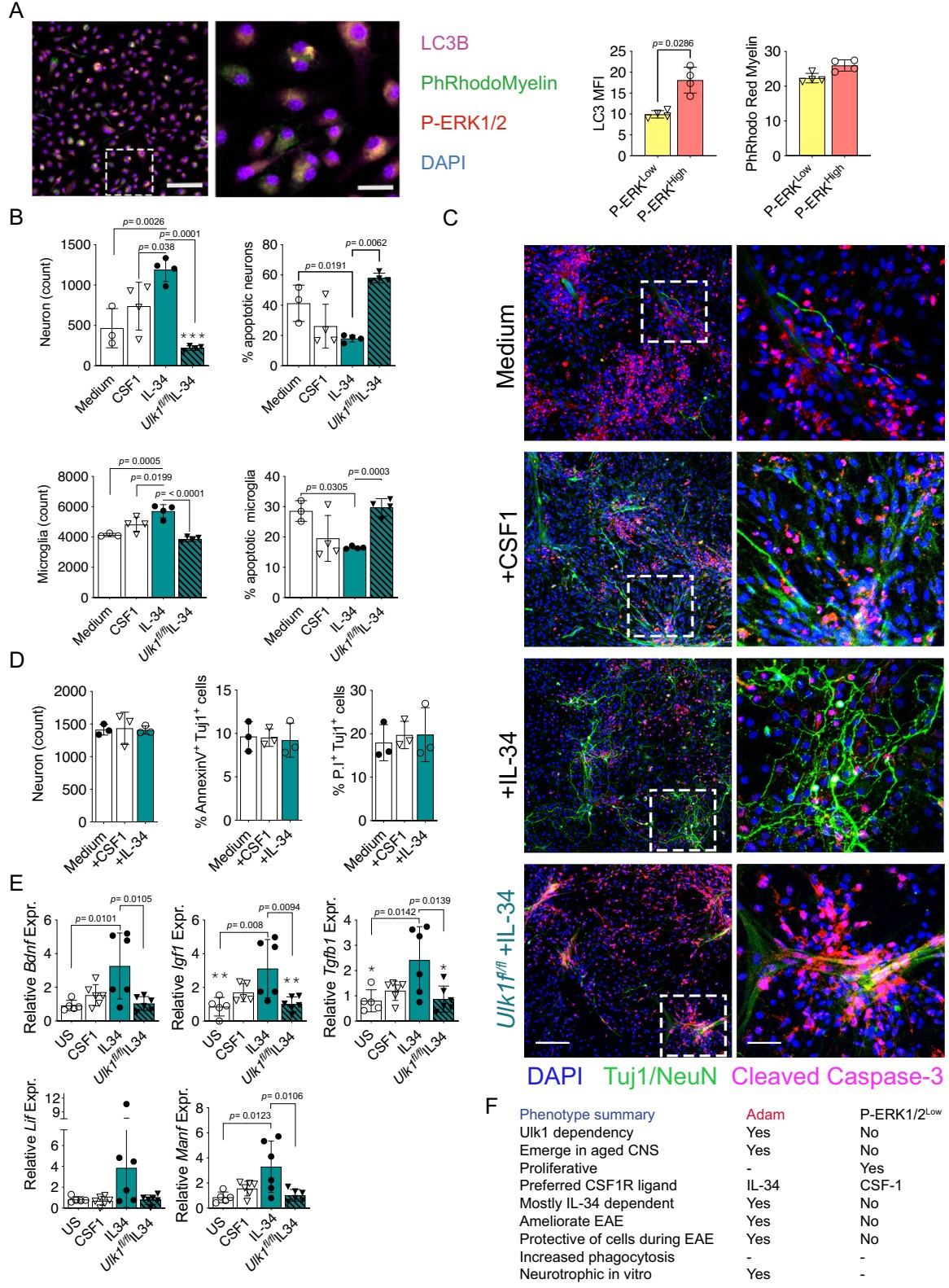

**Flow cytometry**

Naïve and at 7 days post-EAE onset CNS cells were analyzed. Single-cell suspensions were plated and stained with conjugated antibodies and LIVE/DEAD™ Fixable Near-IR Dead Cell Stain (Invitrogen; L34976). Intracellular/Intranuclear staining was performed after permeabilization using a Fixation/Permeabilization kit (BD biosciences/eBioscience). Staining of phosphorylated proteins was performed using fixation buffer (Biolegend; 420801) followed by True-Phos™ (Biolegend; 425401) permeabilization buffer after a brief 20-minute surface staining. The phosphorylation targeted were: ERK1/2-Thr202/Thyr204, AMPK-Thr183/172 and Akt-Ser473.In experiments targeting AnnexinV, a binding buffer was used (Biolegend; 422201). AnnexinV staining and detection of phosphorylated proteins could not be done on the same samples due incompatible buffer requirement. For

**Fig. 8 | IL-34 increase neuronal survival In vitro microglia co-cultures. A** In vitro uptake images and quantification of labeled myelin myelin LC3 MFI by microglial subpopulations from the aged CNS (P-ERK1/2$^{Low}$, $n = 4$; P-ERK1/2$^{High}$, $n = 4$,). Scale bars indicate 250 µm (left) and 25 µm (right). **B** Counts of live and apoptotic (AnnexinV and/or propidium iodide+) microglia and neurons after 72 h co-culture with sorted aged naïve microglia and respective cytokines (Unstimulated, $n = 3$; CSF-1 $n = 4$-6; IL-34, =5; $Ulk1^{fl/fl}$ IL-34, $n = 4$). **C** Immunocytochemistry showing Tuj1/NeuN⁺ neurons and cleaved caspase-3, signaling apoptosis. Conditions as described in **A**. Scale bars correspond to 500 µm (left) and 50 µm (right). **D** Quantified live, apoptotic AnnexinV⁺ and dead Propidium iodide (P.I) Tuj1⁺ neurons after 72 h

stimulation with respective cytokines (all conditions, $n = 3$). **E** mRNA expression of Neurotrophic factors in sorted microglia from co-cultures described in **A** normalized to *Hprt* (Unstimulated, $n = 5$; CSF-1 $n = 6$; IL-34, $n = 6$; $Ulk1^{fl/fl}$ IL-34, $n = 6$ for all targets except *Igf1*, Unstimulated, $n = 5$; CSF-1 $n = 5$; IL-34, $n = 6$; $Ulk1^{fl/fl}$ IL-34, $n = 6$). **F** Summarized "ADAM" phenotypes described in Figs. 2–7 in comparison to the autophagy independent P-ERK1/2$^{Low}$ microglia population. **A** and **E** showing representative data of two independent experiments. **B**–**D** showing representative data of three independent experiments. **A** Mann-Whitney Two-tailed *U*-test. B and **D**, **E** One-way Anova followed by Tukeys or Dunn's post-hoc test. Error bars represent mean + SD.

quantification of newly formed cells, 50 ug EdU (ThermoFischer scientific; C10339, C10340) were administered i.p. at day 1, 3, 5 before sacrifice and processed as described for single-cell suspensions. The cells were stained for surface markers prior to fixation and treated according to the manufacturer's instructions (ThermoFischer scientific; C10424). Cells were acquired using a Gallios flow cytometer (Beckman Coulter) and analyzed using Kaluza software (Beckman Coulter). All antibodies and reagents are specified in the enclosed technical data file. Due to technical obstacles, AnnexinV and EdU staining could not be done simultaneously with phosphorylation quantifications. Information on antibody dilutions is listed in Supplementary data 2.

### Cell sorting and culture

Cells were cultured in Dulbecco's modified Eagles medium (DMEM, Sigma) conditioned with Fetal bovine serum 10% (vol/vol) (FBS, Sigma) and Penicillin/Streptomycin 1% (vol/vol) (Sigma) and M-CSF 20 ng/ml (R&D) or IL-34 (Biolegend). CSF1R was inhibited by PLX3397 10 ng/ml (Selleckchem) and ERK1/2 were inhibited using SCH772984 (Selleckchem) 3 ng/ml. For in vitro assays and transcriptome analysis microglial cells were sorted from single-cell suspensions by Flow cytometry employing the protocol described above or by magnetic beads. By Flow cytometry, cells from mouse CNS were sorted using a BD Influx cell sorter. Microglia were sorted as live CD11b⁺ CD45$^{Intermediate}$ Ly6G⁻ with enhanced yellow fluorescent protein (eYFP⁺) (Fig. S1) followed by P-ERK 1/2 and P-Akt gating (Fig. 3). For bulk microglia sorting in naïve mouse CNS, we used CD11b magnetic beads and column magnetic cell sorter (MACS, Miltenyi Biotec). Neuronal cells for co-cultures were established from cells isolated P5 mouse CNS and cultured with Neurobrew (Miltenyi biotec) for 14 days before microglia were added.

### Nanostring

Microglia were isolated from freshly obtained CNS tissues, and single-cell suspensions were prepared using the Neural Tissue Dissociation Kit T (Miltenyi Biotec). Each sample is composed by CNS from one male and one female. To ensure uniform exposure to buffers and antibodies, cells were subjected to MACS sorting after labeling with CD11b beads (Miltenyi Biotec), counted and plated 500 000 cells per well. For intracellular target antibody labeling, specifically targeting phosphorylated ERK1/2 (P-ERK1/2), cells were permeabilized using a Fixation/Permeabilization kit (BD Biosciences/eBioscience). The separation of P-ERK1/2 high and low cells was achieved using a BD Influx cell sorter, with microglia identified through gating as YFP+, CD45$^{Intermediate}$, and Ly6G-. Throughout the processing, both tissues and cells were maintained at a low temperature to ensure optimal conditions. P-ERK1/2$^{High}$ and P-ERK1/2$^{Low}$ microglia from aged mice were compared by transcriptome analysis internally and to microglia from adult CNS. For this purpose, we used the Nanostring nCounter system employing the "Neuroinflammation" panel of 770 genes and the nSolver software. This method has been shown before to be suitable for the analysis of RNA in cells undergone the fixation protocol required for flow cell sorting based on targeted phosphorylations[108]. GSEA analysis performed with standard settings–classic scoring

scheme for the enrichment score signal-to-noise metrics for the ranked gene list and presented as upregulated genes/gene set covered by the panel. The Supplementary data 1 contains all raw and normalized Nanostring nCounter expression data.

### Immunocytochemistry

CNS cells were sorted and plated into poly-L-lysine coated plates and incubated for 36 h before adding purified myelin for an additional 12 h. After washing, with 0.2% Tween-20. Non-specific binding was blocked by adding 10% BSA and serum from secondary antibody-producing species. Cells were then incubated overnight with primary antibodies diluted in PBS containing 1% BSA and 0.2% Tween-20. After washing, secondary antibodies diluted in host serum were added and incubated at 37 °C for 1 h. Finally, DAPI solution (4′,6-Diamidino-2-Phenylindole, Dihydrochloride, 0.2 µg/ml, BD Biosciences) was added to the wells for 3 min before final washing. Samples were analyzed using Zeiss LSM880 microscope and Zeiss Zen software. Cellprofiler™ (Broad institute) software was used for quantitative analyses. Images were acquired using the same settings for all samples. P-ERK1/2 high and low populations were discriminated by applying a treshhold in Cellprofiler™ with a qualitative control of visual appearance. The threshold was slightly adjusted between experimental analysis but not within the same panels. Primary antibodies used were anti-Cleaved Caspase 3 (9661 S, Cell signaling technologies), anti- NeuN (MAB377, Sigma-Aldrich, concentration 1:250 in PBS + 1% BSA and 0.2% Tween-20), anti-Tuj1 (801202, BioLegend, concentration 1:500 in PBS + 1% BSA and 0.2% Tween-20), CHOP (1335, Novus Biologicals, concentration 1:250 PBS PBS + 1% BSA and 0.2% Tween-20), P-ERK1/2 (369501, Biolegend, concentration 1:500 in PBS + 1% BSA and 0.2% Tween-20) LC3B (NB600-1384, Novus biologicals, concentration 1:500 in PBS + 1% BSA and 0.2% Tween-20) and secondary antibodies diluted 1:500 in PBS + 2,5% host serum.

### Immunohistochemistry of naïve CNS

Perfused brain and spinal cord tissues were immersed and fixed in 4% PFA for 24 h, followed by sucrose protection treatment (20%) for at least 48 h. Tissues were then sectioned at 14 µm after being divided and embedded in OCT cryomount (Histolab) and frozen in isopentane. Sections were then incubated in 10% FCS /0,05% Triton X-100 in PBS (blocking solution) for 30 minutes followed by staining with primary antibodies specific to P2RY12 (S16007D, BioLegend, concentration 1:500 in PBS + 0,3% Triton X) and P-ERK1/2 (6B8B69, BioLegend, concentration 1:500 in PBS + 0,3% Triton X) and LC3 (ab48394, Abcam, concentration 1:500 in PBS + 0,3% Triton X) and incubated overnight. The next day sections were stained with secondary antibodies listed in the technical sheets. Confocal images were acquired using a Zeiss LSM880 microscope and Zeiss Zen software.

### Histopathology and immunofluorescence of EAE CNS

For histopathology, paraffin embedded brain and spinal cord cross-sections (3–5 µm thick) were dewaxed in xylol, rehydrated and then stained with Hematoxylin & Eosin (HE) and Luxol Fast Blue (Klüever) to assess tissue inflammation and demyelination, respectively. The inflammatory index (I.I.) and demyelination score (DM) were

determined by analyzing the mouse brain- and spinal cord cross-sections as previously described using an Olympus light microscope[109].

For immunofluorescence (IF), 12 μm thick cryosections were obtained in the cryostat and processed as following. The sections were first air-dried for 30 minutes and the washed 3 times in PBS. Subsequent antigen retrieval step was performed in citrate buffer (pH 6.0) for 40 minutes in a steamer device (Braun, Germany). Sections were then incubated in 10% FCS /0,05% Triton X-100 in PBS (blocking solution) for 30 minutes, followed by incubation overnight at 4 °C with primary antibodies diluted in the blocking solution. Target-specific primary antibodies used for IF were rat anti-mouse CD68 (MCA1957, Biorad, concentration 1:500 in PBS + 10% BSA and 0,05% Triton X) and mouse anti-mouse amyloid precursor protein (APP; MAB348, Sigma-Aldrich, concentration 1:500 in PBS + 10% BSA and 0,05% Triton X). Secondary antibodies (donkey anti-rat and donkey anti-mouse) conjugated with Alexa Fluor dyes were obtained from Invitrogen. DAPI (4′,6-Diamidino-2-Phenylindole, Dihydrochloride, 0.2 μg/ml) was included in the last PBS washing step to visualize the nuclei. Finally, the sections were mounted using ProLong Gold Antifade reagent (Molecular Probes). Quantification of the CD3+ cells is represented as the number of the positive cells per mm2. For APP and CD68+ cells quantification, respectively, 4 confocal z-stacks per animal/brain were acquired, and the area of immunoreactivity above a set threshold was determined using ImageJ64. All images were acquired with a Zeiss LSM880 confocal microscope using ZEN software (Carl Zeiss Microimaging GmbH). The histopathological and IF analyses were performed by two blinded evaluators, respectively.

## qRT−PCR
CNS tissue and cells sorted from CNS or from in vitro cultures were lysed in RLT buffer (Qiagen) and homogenized using Qiashredder (Qiagen). RNA was extracted using RNeasy mini kit (Qiagen) and cDNA synthesized with an iScript cDNA Synthesis Kit (Bio-Rad) according to manufacturers provided protocol. qRT−PCR was performed by quantification of SYBR green (Bio-Rad) and analyzed using the Biorad CFX manager. Expression was normalized to *Hprt* or *Hprt* and *B-actin*. All primer sequences are listed in Supplementary data 2.

## Statistical analysis
GraphPad Prism 10 (http://www.graphpad.com/) was used for all the statistical analysis. All figure legends include information regarding statistical tests used and sample size. No data-points were excluded. In our experimental analyses, we employed both Anova (Analysis of Variance) and t-tests exclusively for datasets that exhibited normal distribution, as confirmed by the Shapiro-Wilk test with a significance level set at 0.05. Notably, throughout our figures and legends all p-values are shown. Detailed statistical information, including p-values and group comparisons, can be referenced in the Source data file.

## Reporting summary
Further information on research design is available in the Nature Portfolio Reporting Summary linked to this article.

## Data availability
All data related to figures are enclosed as supplementary files. The Nanostring expression data generated in this study have been deposited in the NCBI GEO database under accession code GSE248184. Source data are provided with this paper regarding all figures in the Source data file and the Nanostring expression data as Supplementary data 1. Source data are provided with this paper.

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

## Acknowledgements

The authors would like to thank the staff at the animal facility at Karolinska University Hospital and in particular Helen Kungsmark for animal care taking. The project was supported by grants from the Swedish Research Council, the Swedish Brain Foundation, Margaretha af Ugglas foundation, Stockholm County Council (ALF), Svenska sällskapet för medicinsk forskning (SSMF, grant agreement No PD21-0174), Neuroförbundet, Chinese Scholarship Council (CSC), Knut and Alice Wallenberg Foundation and the European Research Council (ERC) under the European Union's Horizon 2020 research and innovation programme (grant agreement No 818170).

## Author contributions

The study was conceived by R.B., A.O.G.C., T.O. and M.J. Most experiments were conducted by R.B. with assistance from Y.C., E.P. and A.O.G.C. M.Z.A. and M.Z. performed histopathology assessments. R.B. and A.O.G.C. wrote the manuscript with input from M.J. The project was supervised by A.O.G.C, T.O. and M.J.

## Funding

## Competing interests

The authors declare no competing interests.
