## [Peer Review File · Nature Communications]

The aging mouse CNS is protected by an autophagy-dependent microglia population promoted by IL-34REVIEWER COMMENTS

Reviewer #1 (Remarks to the Author):

This is an interesting and very well performed study, describing aging effects in microglia and their functional relevance in a model of brain inflammation. The authors show that aging leads to an expanded microglia cell population with activated autophagy, which is associated with an altered intracellular signaling following CSF1R activation. This microglia alteration is also reflected by a pro-inflammatory activation phenotype. This microglia phenotype is reduced in animals with Ulk1 deficiency. When this is done in animals with autoimmune encephalomyelitis, disease is augmented and this is associated with increased cell death of neurons and oligodendrocytes. This microglia alteration could be counteracted in part by intrathecal application of IL34.

The study is very well performed and the documentation of the results is largely convincing. Overall, this is an important novel finding on age dependent microglia changes, which appear to be relevant for a variety of different inflammatory conditions of the brain.

There are some concerns regarding the experiments in EAE models:

a) The authors claim a direct relevance for multiple sclerosis. However, what the data show are effects just in the model of EAE, which has its limitations as a model for MS. The authors should just claim relevance for brain inflammation in general, but not for MS, as long as they do not provide additional evidence shown in MS patients.

b) It is unfortunate that the clinical disease and pathology of the EAE experiments is not characterized in detail. There are a number of questions open: What is the exact difference in clinical disease and pathology of EAE in young versus aged animals in this particular model, to what extent does Ulk1 deficiency change the basic pathology. Do the animals have more extensive demyelination, larger demyelinated lesions, less remyelination? Do aged animals have a chronic progressive disease course in comparison to young animals?

c) The mechanisms, how IL34 reduces damage is unclear. Although the authors describe that this is associated with some increased expression of some cytokines (e.g. Bdnf, Lif, Igf1), the actual relation to the degenerative phenotype is not fully clear and direct proof of the involvement of these cytokines is absent.

Minor points: Introduction line 55: The authors argue that in adult microglia depletion models the cells are replaced by recruited myeloid cells, while after pharmacological depletion they are repopulated by local proliferation. This distinction is not true, since also in adult microglia depletion they can be replaced by proliferating microglia (Rubino et al Nat Commun).

Reviewer #2 (Remarks to the Author):

Here, Berglund et al studied how microglia states change during aging, demonstrating the appearance of age-dependent microglia states with distinct phenotypic features. By leveraging a gene dataset from transcriptome analysis of aging-related p-ERK^{high} microglia, the authors identified activated autophagy pathway. Of note, Ulk1 deletion impaired the appearance of the age-dependent microglia. In addition, using EAE model, the authors tested the effect of Ulk1 deletion in microglia, and found a protective role of Ulk1-dependent microglia state, which they claimed depends on IL34 signaling. Overall, this study is interesting and provides novel aspects of how microglial states are regulated through ULK1. The paper is clearly written, and the abstract is appropriate. However, despite the significant relevance of the topic the referee's enthusiasm is strongly dampened by several major points that limit the value of the study. For example, for the quantification with cell-type specific Ulk1 deletion samples, the authors used aged wildtype mice as a control to compare with Ulk1-deficient (Cx3cr1-CreERT2;Ulk1^{fl/fl}). However, Cx3cr1-CreERT2 heterozygotes express less CX3CR1 in microglia, which is known to affect several aspects of microglial properties. This may also be the case for their phenotype during aging (e.g. Fig.4H),

thus Cx3cr1-CreERT2 hetero mice should be included as a proper control. Furthermore, the separation of microglia and BMDM by CD45 expression doesn't work properly especially when using disease models (e.g. EAE), because microglia upregulate CD45 in such context. Therefore, FACS-based quantification of microglia and BMDM is not reliable, which needs alternative gating strategies.

Additional points and suggestions

1. In which brain areas are shown in Fig 1H? Are there any regional differences in Fig1A-I? The quantification in distinct brain regions (e.g. gray vs. white matter) would be informative, since the authors claimed CSF1R ligands Csf1/IL34 play distinct role for the appearance of age-related microglia phenotype.
2. Cx3cr1-mediated gene targeting is not specific for microglia, but other CNS resident myeloid cells like BAM/CAM (PMID: 32541832).
3. Fig.5 is confusing. IL-34 treatment ex vivo doesn't induce either p-ERK in microglia (Fig.5D) or reactive microglia (Fig.5F). How do the authors say that IL-34 is an important mediator for the ADMA population?
4. The "ADAM" population during aging and in EAE is comparing in terms of gene expression? If not, it can be highly misleading. ADAM in both contexts should be profiled.
5. What about Csf1 and Il34 levels during EAE?

Minor

1. Fig.3G is basically validation of Fig.2 dataset, which should be combined.

Reviewer #3 (Remarks to the Author):

The article by Berglund et al. describes a new microglia population or "signature" that protects the aged brain in front of neuroinflammatory challenges such as experimental autoimmune encephalomyelitis, a mouse model of multiple sclerosis. They name this microglial population as ADAM (Autophagy Dependent Age-acquired Microglia).

This microglial population is characterized by increased phosphorylation of ERK and AMPK, and enhanced expression of autophagy and lysosome-related transcriptome. In line with this, the deletion of the autophagy pre-initiation complex member ULK1 conditionally in microglia in aged mice ablates this population and its protection in front of neuroinflammatory disease. They also describe that aging microglia exhibits altered CSF1/IL34-CSF1R axis signaling, and IL34 treatment (but not CSF1) selectively expands ADAM and protects the brain from EAE in aged mice.

Overall, the scientific soundness of the data presented in the manuscript is good, and the conclusions are in line with the data presented, albeit some of them are supported by indirect evidence.

The STRENGTH of the article is the conceptual innovation on microglia and multiple sclerosis field. Indeed, a new microglia population is described in the aging brain, with a possibility of pharmacological modulation (activation of the IL34-CSF1R axis).

The main WEAKNESS of the article in my opinion is the lack of evidence of real autophagy-dependency of this microglia population (only indirect transcriptomic evidence is provided on the possible activation of autophagy in microglia).

Autophagy is a dynamic and complex process and the up-regulation of upstream regulators such as phosphorylated AMPK and some ATG or lysosomal proteins, does not necessarily mean that autophagy flux is activated. Indeed, the characterization of the autophagic response needs to be made by complementary techniques in order to understand the whole autophagosome biogenesis and clearance response and its regulation. Moreover, as stated by the authors, autophagy proteins also participate in other endo/phago-lysosomal related networks and functions, and the

interactions between autophagy, endocytosis, and phagocytosis are highly complex. The message of the manuscript would be more powerful if authors provided data on the real autophagic status of ADAM. Many assertions made in the discussion section are based on indirect data of the autophagic status of these cells.

Other concerns include:

- Given that IL34 is a focus molecule on the main scientific message of the manuscript, it would be interesting to include the existence of receptors other than CSF1R that respond to IL34 in the introduction section.

- Some references have not been updated: reference #39 was retracted, reference #55 was published in 2021.

- Line 80: when talking about autophagy ..."recycling, including surface receptors"... Line 83: "findings by us and others have expanded the function of autophagy... recirculation of surface receptors"...

The authors may want to re-phrase for accuracy.

- The material and methods section should be refined.

On experimental subjects: it is of high relevance for this article to include the schedules to delete ULK1 with tamoxifen injection in mice, describing the time points of tamoxifen injection and the experimental time-points for the different ages tested (then, the reader can know how exactly the experiment was performed, and for how long ULK1 was absent in microglia).

Line 449: intracisternally: which cisterna?

Line 469: mice were fed with wet food FOR... (reason is missing).

Two independent sections have been included for cell cultures; they should be described consecutively in the same section in my opinion.

Lines 517-519: reference/s?

Immunofluorescence cell and histological studies AND Immunohistochemistry and image analysis: the titles are not accurate; these sections could be combined or clearly specify what they mean for. Line 534: secondary antibodies (which?)

Statistical analysis: the authors should include a clear description of their statistical analysis strategies, including analysis of normality, equality of variances, selection of parametric and non-parametric analysis. ANOVA is always one-way, why did the authors select this type of analysis, for example, when the experimental design contains two factors (for example, Figure 1J and others)? Which was the established level of significance?

- Line 116: ..."an altered response to CSF1R activation"... In my opinion, the authors describe an alteration of the whole CSF1/IL34-CSF1R axis. This should be taken into account when concluding the section (lines 154-155).

- Line 147: I suggest modifying pharmaceutical for pharmacological inhibition

- Lines 161-163: I suggest explicitly indicating that the comparison made was between adult, aged pERK low, and aged pERK high and refer to the reason (figure1).

- Lines 180-182: I think it would help referring to Figure 2B too.

- Line 186: ..."enhanced expression of cd47"... As I interpret the graph, its expression would be down-regulated.

- Line 238: those results are shown in Figure4E-F? Please, specify the schedule of ULK1 deletion on Figure 4H. It is not clear which exact interventions are being compared in this experiment. Figure 4I: is this ULK1 fl/fl TAM > 24m?,

- Line 250: ..."this autophagy-dependent population was found to be less proliferative"... than what, which other population? It is not clear to me which is the comparison.

- Figure 5F: why is IL-34 effect described as reactive when the levels of the markers are the same as in PBS treated group?

- Line 298: ..."remaining microglia, even when expanded, failed to protect"... Figure6C, the turnover of microglia was similar, was the population expanded?

- Figure 7A: was survival not assessed in this experiment?

- Figure 7C: Statistical comparison between PBS and IL34 receiving groups was not performed for each genotype?

Point-by-point response response to reviewer comments regarding the manuscript “*The aging CNS is protected by an autophagy-dependent microglia population promoted by IL-34*”.

Red indicates new text.

Blue indicates comments in this letter

Black is the original text or comment.

Reviewer #1 (Remarks to the Author):

This is an interesting and very well performed study, describing aging effects in microglia and their functional relevance in a model of brain inflammation. The authors show that aging leads to an expanded microglia cell population with activated autophagy, which is associated with an altered intracellular signaling following CSF1R activation. This microglia alteration is also reflected by a pro-inflammatory activation phenotype. This microglia phenotype is reduced in animals with Ulk1 deficiency. When this is done in animals with autoimmune encephalomyelitis, disease is augmented and this is associated with increased cell death of neurons and oligodendrocytes. This microglia alteration could be counteracted in part by intrathecal application of Il34.

The study is very well performed and the documentation of the results is largely convincing. Overall, this is an important novel finding on age dependent microglia changes, which appear to be relevant for a variety of different inflammatory conditions of the brain.

Response: We appreciate Reviewer’s overall positive evaluation of our study and the relevance of the findings as well as the input that has considerably improved our interpretations.

There are some concerns regarding the experiments in EAE models:

a) The authors claim a direct relevance for multiple sclerosis. However, what the data show are effects just in the model of EAE, which has its limitations as a model for MS. The authors should just claim relevance for brain inflammation in general, but not for MS, as long as they do not provide additional evidence shown in MS patients.

Response: We agree with the Reviewer that the translation of findings from EAE to MS is not trivial. We have already made a statement towards the model and its limitations (lines 486-489: “Although the EAE model has limitations in translation to human MS disease,”).

We also often state throughout the text that the model reflects “MS-like disease” and/or “neuroinflammation”. We consider these statements nuanced in context of the facts that many findings in EAE replicate findings in human MS and the vast majority of MS therapies are effective in EAE, as reviewed by Robinson et al. ¹. Nevertheless, we believe that it is essential to clarify that our study does not claim to be an MS investigation but rather an exploration of pathogenic immunological mechanisms relevant to the central nervous system (CNS) diseases, that could also be of relevance for MS. The alterations in the CSF1R axis have also been observed by other researchers in the context of MS. For instance, a study by Hagan et al. revealed increased CSF1 expression in tissue samples from both relapsing and progressive MS patients, along with elevated CSF1R expression and CSF1 protein levels in

the cerebrospinal fluid (CSF). Importantly, IL-34 levels remained unaffected. This indicates a polarization towards CSF1 induced microglial states over IL-34².

b) It is unfortunate that the clinical disease and pathology of the EAE experiments is not characterized in detail. There are a number of questions open: What is the exact difference in clinical disease and pathology of EAE in young versus aged animals in this particular model, to what extent does Ulk1 deficiency change the basic pathology. Do the animals have more extensive demyelination, larger demyelinated lesions, less remyelination? Do aged animals have a chronic progressive disease course in comparison to young animals?

Response: We thank the Reviewer for highlighting this issue. We have conducted a comprehensive assessment of the CNS during EAE and have in the aged mice with microglia deficient in ULK1 observed pathological changes predominantly concentrated within the brain. These changes include significant instances of demyelination, infiltration of macrophages, and the presence of Amyloid Precursor Protein (APP) deposits, which serve as markers of pathological processes. These experimental findings have been incorporated into Fig. 6 and Fig. S2 (as shown below) and corresponding text has been added to the Results and Discussion section. The aged mice had in comparison to adults aggravated demyelination of the spinal cord but no indications of more pathological features in the brain. We have in a previous publication shown a more chronic EAE in aged mice versus adults³ but did not repeat an experiment with a full clinical course within this project. Remyelination is notoriously difficult to study but an interesting question that needs its own designated study in relation the microglial loss. We did however study myelin phagocytosis which is essential for remyelination and found the P-ERK1/2 high microglia not to differ from their low counterparts in regard of myelin debris uptake (as shown below).

Added text to results:

“Microglial phagocytosis plays a pivotal role in mitigating central nervous system (CNS) damage across various pathological conditions, including experimental autoimmune encephalomyelitis (EAE)⁸²⁻⁸⁴. In our investigation, we observed an elevated surface density and increased expression of scavenger receptors within the ADAM population, as illustrated in Figure 2.

Hence, we aimed to ascertain whether this phagocytic capability underwent alterations in the ADAM population. To do so, we conducted an ex vivo experiment, exposing microglia from aged mice to fluorophore-conjugated myelin debris. Surprisingly, our analysis revealed a non-significant difference in the uptake of myelin debris when comparing the P-ERK1/2^{High} and P-ERK1/2^{Low} subsets although the autophagosome marker LC3 was detected at higher density in ADAM (Fig. 8a). This observation suggests the existence of additional neuroprotective functions within the ADAM population. So to address the mechanism behind the neuroprotection following IL-34 treatment, we co-cultured neurons with microglia sorted from aged Wt and Ulk1^{fl/fl} mice and exposed them to CSF-1 or IL-34 or unstimulated control. IL-34 treatment increased neuron counts and reduced the number of apoptotic neurons when co-cultured with Wt microglia but not Ulk1-deficient microglia (Fig. 8b-d)”.

And this section:

*“Immunohistochemical analysis unveiled a pronounced augmentation in the demyelinated lesion load in *Ulk1^{fl/fl}* mice lacking the ADAM population (Fig. 6e-h). These demyelinated regions exhibited a heightened prevalence of CD68⁺ macrophages, presumably originating from monocytes (Fig. 6f). Additionally, conspicuous deposits of amyloid precursor protein (APP) were evident within the tissue, indicative of neuronal pathology⁸¹ (Fig. 6h). Notably, these findings manifested primarily within the cerebral tissues of *Ulk1^{fl/fl}* mice, while the spinal cord exhibited an absence of pathological alterations. This spatial discrepancy is congruent with the divergent influence exerted by IL-34 versus CSF-1, and aligns with the lethal EAE phenotype detected. CNS of the *Ulk1^{fl/fl}* mice also exhibited extensive macrophage activation in tissue outside demyelinating lesions, suggesting a repopulation of previously vacant niches (Fig. S2b). Throughout the course of EAE, we observed a significant increase in the expression of *Csf1* but not *Il34*. This finding underscores the widely recognized inflammatory responsiveness associated with *Csf1* (Fig. S2c).*

In conclusion, naïve ADAM-deficient mice did not show reduced counts of neurons or cells of the oligodendrocyte lineage. However, when challenged with EAE the remaining microglia failed to protect the CNS cells from death and EAE-associated mortality”.

Added text to discussion

*“The pathology observed during EAE predominantly affected the brain instead of the spinal cord, a phenotype rarely seen in EAE. This observation supports the concept of a protective microglial phenotype associated with IL-34, which is primarily expressed in cortical brain regions. Knowing that microglia are the main targets of IL-34 we believe the protective effect to be secondary to expansion of the ADAM population, a statement supported by the absent effect in mice with *Ulk1* deficiency. We could further validate a neuroprotective phenotype of the IL-34 treated microglia in vitro by cell viability measurements, possibly explained by increased expression of neurotrophic factors. Notably we did not detect an increased increase in phagocytic capacity of the ADAM microglia, a function usually associated with DAM phenotypes and health promoting capacities of microglia in general. These experiments were however measuring specifically myelin uptake and lysosomal loading which not exclude altered phagocytosis of e.g. apoptotic neurons”.*

In vitro myelin phagocytosis

c) The mechanisms, how Il34 reduces damage is unclear. Although the authors describe that this is associated with some increased expression of some cytokines (e.g. Bdnf, Lif, Igf1), the

actual relation to the degenerative phenotype is not fully clear and direct proof of the involvement of these cytokines is absent.

Response: Our message underscores that IL-34 plays a pivotal role in promoting the expansion and polarization of a protective microglia phenotype. While we acknowledge that the assessment of neurotrophic factor expression offers only a superficial characterization, it nonetheless provides valuable indicators. To delve into the precise functions of these cytokines during aging necessitates extensive and time-demanding research, and given the age of study subjects, a task beyond the scope of this particular study.

Minor points: Introduction line 55: The authors argue that in adult microglia depletion models the cells are replaced by recruited myeloid cells, while after pharmacological depletion they are repopulated by local proliferation. This distinction is not true, since also in adult microglia depletion they can be replaced by proliferating microglia (Rubino et al Nat Commun).

Response: We agree and we have adjusted text accordingly (as outlined below):

“The establishment and maintenance of the microglial population is dependent on the activity of CSF1R⁴⁻⁶. In humans, mutations in the CSF1R gene cause severe loss of microglia and associate with lethal abnormalities and degenerative changes in the CNS⁷⁻⁹. In adult microglia depletion models using either genetic targeting or chemical CSF1R inhibition, the niche is rapidly repopulated by bone marrow-derived macrophages (BMDM) and proliferation of residual microglia cells⁹⁻¹¹. BMDM can enter the CNS during inflammation but do not contribute to the myeloid population of the CNS parenchyma when homeostasis is restored^{12, 13}”.

Reviewer #2 (Remarks to the Author):

Here, Berglund et al studied how microglia states change during aging, demonstrating the appearance of age-dependent microglia states with distinct phenotypic features. By leveraging a gene dataset from transcriptome analysis of aging-related p-ERK^{high} microglia, the authors identified activated autophagy pathway. Of note, Ulk1 deletion impaired the appearance of the age-dependent microglia. In addition, using EAE model, the authors tested the effect of Ulk1 deletion in microglia, and found a protective role of Ulk1-dependent microglia state, which they claimed depends on IL34 signaling. Overall, this study is interesting and provides novel aspects of how microglial states are regulated through ULK1. The paper is clearly written, and the abstract is appropriate. However, despite the significant relevance of the topic the referee’s enthusiasm is strongly dampened by several major points that limit the value of the study.

Response: We are grateful for the through scrutiny of our study and we hope that our clarifications and additional data we have addressed the major points of concern raised by the Reviewer.

For example, for the quantification with cell-type specific Ulk1 deletion samples, the authors used aged wildtype mice as a control to compare with Ulk1-deficient (Cx3cr1-

CreERT2;Ulk1^{fl/fl}). However, Cx3cr1-CreERT2 heterozygotes express less CX3CR1 in microglia, which is known to affect several aspects of microglial properties. This may also be the case for their phenotype during aging (e.g. Fig.4H), thus Cx3cr1-CreERT2 hetero mice should be included as a proper control.

Response: We understand the concern, but in fact CX3CR1-Cre heterozygote mice were used in all experiments as stated in M&M as “The experimental Cre mice had a hemizygote *Cx3cr1*^{CreERT} genotype” and in the figure legends as “*Ulk1*^{fl/fl} refers to *Ulk1*^{fl/fl} CX3CR1^{CreERT2} and *Wt* to *Ulk1*^{wt/wt} CX3CR1^{CreERT2}.”

Furthermore, the separation of microglia and BMDM by CD45 expression doesn't work properly especially when using disease models (e.g. EAE), because microglia upregulate CD45 in such context. Therefore, FACS-based quantification of microglia and BMDM is not reliable, which needs alternative gating strategies.

Response: While the gating strategy we employ is well-established, it has occasionally been a subject of debate. To provide further clarity on this matter, we have included Fig. S2, which showcases MFI plots for LY6C, a marker highly expressed in monocytes, as well as CX3CR1 (YFP) and CD11c, markers more predominant in microglia and other resident macrophage populations (as illustrated below). This illustrates a gating strategy where we with the antibody clones and conjugates we use can separate microglia from infiltrating bone marrow derived cells to a high degree.

Flow cytometry analysis of CNS myeloid cells during EAE

Additional points and suggestions

1. In which brain areas are shown in Fig 1H? Are there any regional differences in Fig 1A-I? The quantification in distinct brain regions (e.g. gray vs. white matter) would be informative, since the authors claimed CSF1R ligands Csf1/IL34 play distinct role for the appearance of age-related microglia phenotype.

Response: We appreciate Reviewer's attention to this significant matter. In Figure 4, we have now incorporated experimental data illustrating that the population characterized by high levels of P-ERK1/2 predominantly resides within the cortical regions of the brain. This finding aligns with earlier research, consistent with the distinctions between IL-34 and CSF-1 (as shown below) and the corresponding text has been added to Result and Discussion sections sections and reads as follows:

Added to Results:

“Immunohistochemical analysis revealed a notable decrease of the ADAM population, predominantly in the grey matter regions (Fig. 4g, h). Within these regions, we observed a significant upregulation of LC3 in the P-ERK1/2^{High} population in Wt mice, which may suggest the initiation of autophagosome formation (Fig. 4i). These findings align with earlier studies that associate IL-34 with microglia in grey matter and CSF-1 with those in white matter^{14, 15}. Notably, since the microglial population count was not affected in aged mice when Tamoxifen-induced deletion of Ulk1 was done one month before they were sacrificed (Fig. 4e, f), we concluded that the reduction of the microglia, particularly ADAM, population was caused by a long-term cumulative loss. Concurrent with the loss of microglia, the expression of Csf1 and Il34 in the CNS was decreased (Fig. 4j)^{16, 17}”.

Added to Discussion:

“While CSF-1 associates with inflammation and is secreted mainly by microglia and other immune cells, IL-34 is sourced by neurons and glial cells and is more abundant in steady-state but is yet to be comprehensively explored in neuroinflammatory disease. Studies disrupting IL-34 and CSF-1 gene expression or treatment with specific blocking antibodies suggest regional dependency for these cytokines, with IL-34 affecting mostly the grey matter^{15, 18}. Accordingly, the observed absence of microglia was primarily detected in the cortical areas of the brain, an observation that potentially could elucidate the underlying factors contributing to the observed mortality.”

Immunohistochemical analysis of naïve CNS from aged mice

2. Cx3cr1-mediated gene targeting is not specific for microglia, but other CNS resident myeloid cells like BAM/CAM (PMID: 32541832).

Response: Although CX3CR1-mediated gene targeting represents a valuable approach, it does not, as the reviewer points out, achieve complete specificity towards microglia, potentially impacting other CNS resident myeloid cells known as brain-associated macrophages (BAM) or CNS-associated macrophages (CAM). It is noteworthy that BAM/CAMs are, to varying degrees, repopulated from bone marrow-derived monocytes. In our experiments, these monocytes will not be affected by *Ulk1* deletion given the time between Tamoxifen treatment and the subsequent experiments, as illustrated in Fig. 5h:

Furthermore, it is important to emphasize that BAMs and CAMs constitute only a small fraction of the overall CNS macrophage population and are typically characterized as CD45 high in experimental settings. In our analysis of naïve brains, the CD45 high population accounts for approximately 10% of the total CD11b+ pool of CNS myeloid cells (as shown below, extracted from data presented in Fig. 3a and 3c for Wt animals). Taken together, the observed clinical impact is highly likely mediated by the impact on microglia. However, if we were to embark on this project today, we would consider employing the microglia-specific HexB cre model for further improved specificity¹⁹.

3. Fig.5 is confusing. IL-34 treatment ex vivo doesn't induce either p-ERK in microglia (Fig.5D) or reactive microglia (Fig.5F). How do the authors say that IL-34 is an important mediator for the ADAM population?

Response: We thank the Reviewer for highlighting the lack of clarity in this section. Our primary finding underscores that IL-34 significantly increases the abundance of the P-ERK1/2 high subpopulation within the aged CNS, as demonstrated in Fig. 5a, b. In contrast, CSF1 does not exhibit a similar effect, as indicated in the same figure. Fig. 5d provides compelling evidence that the elevated levels of P-ERK1/2 in microglia within the aged CNS are sustained by IL-34, while CSF1 appears to have an opposing influence. We have addressed this intriguing observation in the discussion, elaborating on it in line 396.

It is noteworthy that the significance bars in the figures had initially posed some confusion. However, we have since made necessary adjustments to ensure the clarity and accuracy of our data representation (as shown below).

“By intrathecal injections of CSF-1 or IL-34, we expanded the microglia population in aged wild-type mice. While IL-34 expanded the responsive ADAM population and maintained the highly phosphorylated ERK1/2 found in this phenotype, CSF-1 induced the highly proliferative/less reactive population. The exact mechanisms underlying this discrepancy are not addressed in this study but could be influenced by the higher IL-34 affinity to CSF1R, which affects auto-phosphorylation and downstream regulation²⁰. Further, in the aged *Ulk1* deficient mice, IL-34 failed to expand the population establishing support for an IL-34-CSF1R-ERK/AMPK-ULK1 axis”.

Flow cytometry analysis of microglia from naive CNS.

4. The “ADAM” population during aging and in EAE is comparing in terms of gene expression? If not, it can be highly misleading. ADAM in both contexts should be profiled.

Response: We agree with the Reviewer that this is indeed a critical issue. While highly relevant, its worth emphasizing that our categorization of the ADAM population is not solely based on its transcriptional signature. Instead, we consider both its autophagy dependency and the phosphorylation and expression of proteins defining it. Furthermore, it is equally relevant to recognize that the ADAM population in the context of EAE may differ significantly from that in the naive aged CNS. This discrepancy arises from the substantial influence of ongoing inflammation on the transcriptome.

5. What about *Csf1* and *Il34* levels during EAE?

Response: We appreciate that this is brought up and new data is added to Fig. S2 detecting an expected increase in *Csf1* at EAE day 21 post immunization but also a tendency for upregulated *Il34* (as shown below).

pPCR data from CNS 21 days post EAE immunization

Minor

1. Fig.3G is basically validation of Fig.2 dataset, which should be combined.

Response: We thank the Reviewer for this a good suggestion. We have now merged Fig. 3g with Fig. 2.

Reviewer #3 (Remarks to the Author):

The article by Berglund et al. describes a new microglia population or “signature” that protects the aged brain in front of neuroinflammatory challenges such as experimental autoimmune encephalomyelitis, a mouse model of multiple sclerosis. They name this microglial population as ADAM (Autophagy Dependent Age-acquired Microglia).

This microglial population is characterized by increased phosphorylation of ERK and AMPK, and enhanced expression of autophagy and lysosome-related transcriptome. In line with this, the deletion of the autophagy pre-initiation complex member ULK1 conditionally in microglia in aged mice ablates this population and its protection in front of neuroinflammatory disease.

They also describe that aging microglia exhibits altered CSF1/IL34-CSF1R axis signaling, and IL34 treatment (but not CSF1) selectively expands ADAM and protects the brain from EAE in aged mice.

Overall, the scientific soundness of the data presented in the manuscript is good, and the conclusions are in line with the data presented, albeit some of them are supported by indirect evidence.

The STRENGTH of the article is the conceptual innovation on microglia and multiple sclerosis field. Indeed, a new microglia population is described in the aging brain, with a possibility of pharmacological modulation (activation of the IL34-CSF1R axis).

Response: We appreciate Reviewer’s enthusiasm for our findings as well as the depth of the scrutiny and input, which has strengthened our findings and interpretations.

The main WEAKNESS of the article in my opinion is the lack of evidence of real autophagy-dependency of this microglia population (only indirect transcriptomic evidence is provided on the possible activation of autophagy in microglia).

Autophagy is a dynamic and complex process and the up-regulation of upstream regulators such as phosphorylated AMPK and some ATG or lysosomal proteins, does not necessarily mean that autophagy flux is activated. Indeed, the characterization of the autophagic response needs to be made by complementary techniques in order to understand the whole autophagosome biogenesis and clearance response and its regulation. Moreover, as stated by the authors, autophagy proteins also participate in other endo/phago-lysosomal related networks and functions, and the interactions between autophagy, endocytosis, and phagocytosis are highly complex.

The message of the manuscript would be more powerful if authors provided data on the real autophagic status of ADAM. Many assertions made in the discussion section are based on indirect data of the autophagic status of these cells.

Response: This is indeed a highly important point. To address it, we have conducted new experiments focusing on the detection of the autophagosome marker LC3 in relation to P-ERK1/2 high (ADAM) and low microglial cells. In these experiments, we observed a higher density of LC3+ structures in ADAM, both in cell cultures and through immunohistochemical analyses of brain tissue ex vivo (as shown below and integrated in Fig. 3, Fig. 8 and Fig. 4).

Given the dependence on the autophagy core gene *Ulk1* and the upregulation of autophagy-related genes, we are confident that this subpopulation exhibits higher autophagy activity and capacity. Nevertheless, we acknowledge and discuss other possible, albeit less likely, explanations in the discussion section (as outlined below).

Immunocytochemistry analysis of microglia cells regarding proportion of P-ERK1/2 high cells and LC3+ structures in these cells (Fig. 3).

Immunohistochemistry analysis of brain and spinal cord (Fig. 4)

Added to Results:

”In order to elucidate the connection between ERK1/2 signaling and autophagy, we conducted *ex vivo* analysis detecting sustained presence of P-ERK1/2^{High} microglia sorted from the aged CNS (Fig. 3a-b). Importantly, the P-ERK^{High} microglia from the aged CNS had higher autophagosome density, as defined by the presence of LC3B-positive structures (Fig. 3c).

These findings, together with the link between preferential AMPK and ERK1/2 activation downstream of CSF1R and the transcriptional profile, strongly suggest elevated autophagy activity of aged microglia. To evaluate this we targeted *Ulk1*, a gene which encodes a protein required for induction of canonical autophagy, in mice expressing Cre recombinase under the *Cx3cr1* promoter to create an inducible microglia *Ulk1* deletion upon tamoxifen treatment (the strain *Ulk1^{fl/fl}CX3CR1^{CreERT2}* hereafter referred to as *Ulk1^{fl/fl}*)²¹”

Added to discussion:

“This changed phosphorylation pattern is highly indicative of increased canonical autophagy. Several lines of evidence, including the known regulatory functions of detected phosphorylation patterns together with validation of autophagy activation by transcriptional analysis and *ex vivo* correlation between elevated P-ERK1/2 levels and increased LC3B⁺

compartment density, suggests enhanced autophagy activation, although we cannot exclude additional influence of reduced lysosomal degradation.”

Other concerns include: Text issues etc that will be addressed.

- Given that IL34 is a focus molecule on the main scientific message of the manuscript, it would be interesting to include the existence of receptors other than CSF1R that respond to IL34 in the introduction section.

Response: This concern is valid and addressed in the discussion (as outlined below). We do, however, believe the findings are CSF1R driven since inhibition of this receptor abolishes the phenotype as shown in Fig. 1 (as shown below)

Lines 444-448;

“While CSF-1 exclusively engages the CSF1R receptor, IL-34 exhibits binding affinity towards two additional identified receptors: phosphatase- ζ (PTP- ζ) and syndecan-1. The implications of these interactions on microglial function warrant further investigation but is not addressed further in this study²²”.

Flow cytometry analysis of microglia (Fig. 1)

- Some references have not been updated: reference #39 was retracted, reference #55 was published in 2021.

Response: We thank the Reviewer for pointing up these errors. Reference 39 has been removed and 55 updated.

- Line 80: when talking about autophagy ...”recycling, including surface receptors”... Line 83: “findings by us and others have expanded the function of autophagy... recirculation of surface receptors”...

The authors may want to re-phrase for accuracy.

Response: We thank the Reviewer for bringing this to our attention. This is now addressed (as outlined below):

Lines 81-85:

”By altering cellular components for recycling, including surface receptors and metabolic organelles, autophagy appears as an integrative regulator of immune cells, hence a target for

*pharmaceutical immunomodulation*²³. In recent years, findings by us and others have expanded the function of autophagy and its associated proteins to other roles, including phagocytosis and MHC class II presentation²⁴⁻²⁶”

- The material and methods section should be refined.

On experimental subjects: it is of high relevance for this article to include the schedules to delete ULK1 with tamoxifen injection in mice, describing the time points of tamoxifen injection and the experimental time-points for the different ages tested (then, the reader can know how exactly the experiment was performed, and for how long ULK1 was absent in microglia).

Response: We have added information to the M&M section and to the schematic cartoon in Fig. 5 (as shown below). Conditions are also described in legends.

Lines 506-510:

“Aged mice refer 20-25 months old and Adult mice 3-5 to months old. The experimental Cre mice had a hemizygote $Cx3cr1^{CreERT}$ genotype. Tamoxifen (TAM; Sigma), 4mg dissolved in corn oil was administrated subcutaneously three times in 48-hour intervals in 1 to 3 month old mice except in figure 3 were mice older than 20 months had Tamoxifen treatment 1 month prior the experiment.”

Fig. 5:

Line 449: intracisternally: which cisterna?

Response: The compounds were injected to cisterna magna referred to as intracisternatly, we decided however to change to “intrathecally” which is the more common nomenclature.

Line 469: mice were fed with wet food FOR... (reason is missing).

Response: We thank the Reviewer for bringing this to our attention. We have changed text that now reads as follows:

“Following the onset of the disease, mice were provided with wet food due to their compromised ability to forage for both food and water, due to motor impairments.”

Two independent sections have been included for cell cultures; they should be described consecutively in the same section in my opinion.

Response: The sections have now been merged.

Lines 517-519: reference/s?

Response: The relevant references have been added.

Immunofluorescence cell and histological studies AND Immunohistochemistry and image analysis: the titles are not accurate; these sections could be combined or clearly specify what they mean for.

Response: We have now changed the titles.

Line 534: secondary antibodies (which?)

Response: These have all been added to the supplementary file.

Statistical analysis: the authors should include a clear description of their statistical analysis strategies, including analysis of normality, equality of variances, selection of parametric and non-parametric analysis. ANOVA is always one-way, why did the authors select this type of analysis, for example, when the experimental design contains two factors (for example, Figure 1J and others)? Which was the established level of significance?

Response: In our experimental analyses, we employed both ANOVA (Analysis of Variance) and t-tests exclusively for datasets that exhibited normal distribution, as confirmed by the Shapiro-Wilk test with a significance level set at 0.05. Notably, throughout our figures and legends, we have denoted statistical significance using asterisk (*) annotations. Detailed statistical information, including p-values and group comparisons, can be found in our supplementary data file.

Furthermore, it is important to note that dashed lines have been employed to visually separate distinct datasets within the same graph. These datasets were not subjected to joint analysis, and each possesses its own unique context and interpretation.

- Line 116: ...”an altered response to CSF1R activation”... In my opinion, the authors describe an alteration of the whole CSF1/IL34-CSF1R axis. This should be taken into account when concluding the section (lines 154-155).

Response: We agree with the Reviewer and we have adjusted this in the referred section (as outlined below)

”In summary, we found aging microglia to have a reduced turnover and an altered regulatory response downstream of CSF1R in a CNS with age related differences in expression of this receptor and its ligands.”

- Line 147: I suggest modifying pharmaceutical for pharmacological inhibition

Response: Thank you for this suggestion, it has now been changed.

- Lines 161-163: I suggest explicitly indicating that the comparison made was between adult, aged pERK low, and aged pERK high and refer to the reason (figure1).

Response: Thank you for addressing this valid concern. We have now adjusted the text in the result section (as outlined below).

”To characterize the diverged populations, we sorted formaldehyde-fixed adult and aged microglia and further separated the aged microglia based on their ERK1/2 phosphorylation state and analyzed the RNA expression of 770 neuroinflammation-related genes using Nanostring.”

- Lines 180-182: I think it would help referring to Figure 2B too.

Response: This is now addressed in the text.

- Line 186: ...“enhanced expression of cd47”... As I interpret the graph, its expression would be down-regulated.

Response: Cd47 is upregulated in the P-ERK1/2 Low microglia, we have changed the text to clarify this.

”Genes associated with innate immune responses such as Ccl2, Spp1, Cd74 and Cd86 were strongly upregulated in the aged P-ERK1/2^{High} microglia. Meanwhile, P-ERK1/2^{Low} associated partly with homeostatic microglial signatures in MS and the AD animal models and had an higher expression of Cd47 known to inhibit differentiation to a reactive macrophage phenotype (Fig. 2A-B, Suppl. table 1)^{27, 28}”

- Line 238: those results are shown in Figure4E-F? Please, specify the schedule of ULK1

deletion on Figure 4H. It is not clear which exact interventions are being compared in this experiment. Figure 4I: is this ULK1 fl/fl TAM > 24m?,

Response: The *Ulk1* deletion strategy is now clarified in legends, the schematic cartoon in Fig. 5 and in the M&M section (as outlined below). We have changed 24 months to “over 20” (all Aged are in a span of 20-25 months).

“Aged mice refer 20-25 months old and Adult mice 3-5 to months old. The experimental Cre mice had a hemizygote $Cx3cr1^{CreERT}$ genotype. Tamoxifen (TAM; Sigma), 4mg dissolved in corn oil was administrated subcutaneously three times in 48-hour intervals in 1 to 3 month old mice except in figure 3 were mice older than 20 months had Tamoxifen treatment 1 month prior the experiment.”

- Line 250: ...”this autophagy-dependent population was found to be less proliferative”... than what, which other population? It is not clear to me which is the comparison.

Response: We have now clarified it in the text (as outlined below):

”The loss of the ADAM population specifically impacted microglia in the cortical brain regions, and this loss was not mitigated by the expansion of either the remaining microglial cells or infiltrating myeloid populations, while this autophagy-dependent population was found to be less proliferative but with a reactive phenotype.”

- Figure 5F: why is IL-34 effect described as reactive when the levels of the markers are the same as in PBS treated group?

Response: We agree with the Reviewer that this is misleading. We have adressed it in the text (as outlined below):

Lines 295-297:

*”In line with the expansion of the ADAM microglia, we could further show an **maintained** reactive microglia differentiation upon IL-34, **but not CSF-1**, treatment in vivo (Fig. 5F).”*

- Line 298: ...”remaining microglia, even when expanded, failed to protect”... Figure6C, the turnover of microglia was similar, was the population expanded?

Response: We thank the Reviewer for bringing this to our attention, this conclusion actually belongs to the next section. We have now changed the text (as outlined below):

Lines 339-341:

“In conclusion, naïve ADAM-deficient mice did not show reduced counts of neurons or cells of the oligodendrocyte lineage. However, when challenged with EAE the remaining microglia failed to protect the CNS cells from death and EAE-associated mortality.”

- Figure 7A: was survival not assessed in this experiment?

Response: Survival is addressed in Fig 6. In Wt mice we did not detect any mortality.

- Figure 7C: Statistical comparison between PBS and IL34 receiving groups was not performed for each genotype?

Response: Thank you for addressing this matter. We have now conducted statistical analyses, comparing conditions within the same strain and between different strains under varying treatment conditions (as shown below):

Flow cytometry analysis of CNS immune cell populations during EAE.

XXX

1. Robinson AP, Harp CT, Noronha A, Miller SD. The experimental autoimmune encephalomyelitis (EAE) model of MS: utility for understanding disease pathophysiology and treatment. *Handb Clin Neurol.* 2014;122:173-89. doi:10.1016/b978-0-444-52001-2.00008-x
2. Hagan N, Kane JL, Grover D, et al. CSF1R signaling is a regulator of pathogenesis in progressive MS. *Cell Death & Disease.* 2020/10/23 2020;11(10):904. doi:10.1038/s41419-020-03084-7
3. Berglund R, Guerreiro-Cacais AO, Adzemovic MZ, et al. Microglial autophagy-associated phagocytosis is essential for recovery from neuroinflammation. *Sci Immunol.* Oct 16 2020;5(52)doi:10.1126/sciimmunol.abb5077
4. Dai XM, Ryan GR, Hapel AJ, et al. Targeted disruption of the mouse colony-stimulating factor 1 receptor gene results in osteopetrosis, mononuclear phagocyte deficiency, increased primitive progenitor cell frequencies, and reproductive defects. *Blood.* Jan 1 2002;99(1):111-20. doi:10.1182/blood.v99.1.111
5. Stanley ER, Chitu V. CSF-1 receptor signaling in myeloid cells. *Cold Spring Harb Perspect Biol.* Jun 2 2014;6(6)doi:10.1101/cshperspect.a021857
6. Coniglio SJ, Eugenin E, Dobrenis K, et al. Microglial stimulation of glioblastoma invasion involves epidermal growth factor receptor (EGFR) and colony stimulating factor 1 receptor (CSF-1R) signaling. *Mol Med.* May 9 2012;18(1):519-27. doi:10.2119/molmed.2011.00217

7. Rademakers R, Baker M, Nicholson AM, et al. Mutations in the colony stimulating factor 1 receptor (CSF1R) gene cause hereditary diffuse leukoencephalopathy with spheroids. *Nature genetics*. Dec 25 2011;44(2):200-5. doi:10.1038/ng.1027
8. Guo L, Bertola DR, Takanohashi A, et al. Bi-allelic CSF1R Mutations Cause Skeletal Dysplasia of Dysosteosclerosis-Pyle Disease Spectrum and Degenerative Encephalopathy with Brain Malformation. *Am J Hum Genet*. May 2 2019;104(5):925-935. doi:10.1016/j.ajhg.2019.03.004
9. Huang Y, Xu Z, Xiong S, et al. Repopulated microglia are solely derived from the proliferation of residual microglia after acute depletion. *Nat Neurosci*. Apr 2018;21(4):530-540. doi:10.1038/s41593-018-0090-8
10. Lund H, Pieber M, Parsa R, et al. Competitive repopulation of an empty microglial niche yields functionally distinct subsets of microglia-like cells. *Nat Commun*. Nov 19 2018;9(1):4845. doi:10.1038/s41467-018-07295-7
11. Green KN, Crapser JD, Hohsfield LA. To Kill a Microglia: A Case for CSF1R Inhibitors. *Trends in Immunology*. 2020/09/01/ 2020;41(9):771-784. doi:<https://doi.org/10.1016/j.it.2020.07.001>
12. Ajami B, Bennett JL, Krieger C, McNagny KM, Rossi FM. Infiltrating monocytes trigger EAE progression, but do not contribute to the resident microglia pool. *Nat Neurosci*. Jul 31 2011;14(9):1142-9. doi:10.1038/nn.2887
13. Ajami B, Bennett JL, Krieger C, Tetzlaff W, Rossi FM. Local self-renewal can sustain CNS microglia maintenance and function throughout adult life. *Nat Neurosci*. Dec 2007;10(12):1538-43. doi:10.1038/nn2014
14. Kana V, Desland FA, Casanova-Acebes M, et al. CSF-1 controls cerebellar microglia and is required for motor function and social interaction. *J Exp Med*. Oct 7 2019;216(10):2265-2281. doi:10.1084/jem.20182037
15. Easley-Neal C, Foreman O, Sharma N, Zarrin AA, Weimer RM. CSF1R Ligands IL-34 and CSF1 Are Differentially Required for Microglia Development and Maintenance in White and Gray Matter Brain Regions. *Front Immunol*. 2019;10:2199. doi:10.3389/fimmu.2019.02199
16. Zhang Y, Chen K, Sloan SA, et al. An RNA-sequencing transcriptome and splicing database of glia, neurons, and vascular cells of the cerebral cortex. *The Journal of neuroscience : the official journal of the Society for Neuroscience*. Sep 3 2014;34(36):11929-47. doi:10.1523/jneurosci.1860-14.2014
17. Zhang Y, Sloan SA, Clarke LE, et al. Purification and Characterization of Progenitor and Mature Human Astrocytes Reveals Transcriptional and Functional Differences with Mouse. *Neuron*. Jan 6 2016;89(1):37-53. doi:10.1016/j.neuron.2015.11.013
18. Obst J, Simon E, Martin-Estebane M, et al. Inhibition of IL-34 Unveils Tissue-Selectivity and Is Sufficient to Reduce Microglial Proliferation in a Model of Chronic Neurodegeneration. Original Research. *Frontiers in Immunology*. 2020-October-08 2020;11(2360)doi:10.3389/fimmu.2020.579000
19. Masuda T, Amann L, Sankowski R, et al. Novel Hexb-based tools for studying microglia in the CNS. *Nature Immunology*. 2020/07/01 2020;21(7):802-815. doi:10.1038/s41590-020-0707-4
20. Boulakirba S, Pfeifer A, Mhaidly R, et al. IL-34 and CSF-1 display an equivalent macrophage differentiation ability but a different polarization potential. *Sci Rep*. Jan 10 2018;8(1):256. doi:10.1038/s41598-017-18433-4

21. Zachari M, Ganley IG. The mammalian ULK1 complex and autophagy initiation. *Essays Biochem.* Dec 12 2017;61(6):585-596. doi:10.1042/ebc20170021
22. Lelios I, Cansever D, Utz SG, Mildenerberger W, Stifter SA, Greter M. Emerging roles of IL-34 in health and disease. *J Exp Med.* Mar 2 2020;217(3)doi:10.1084/jem.20190290
23. Galluzzi L, Bravo-San Pedro JM, Levine B, Green DR, Kroemer G. Pharmacological modulation of autophagy: therapeutic potential and persisting obstacles. *Nat Rev Drug Discov.* Jul 2017;16(7):487-511. doi:10.1038/nrd.2017.22
24. Heckmann BL, Teubner BJW, Tummers B, et al. LC3-Associated Endocytosis Facilitates beta-Amyloid Clearance and Mitigates Neurodegeneration in Murine Alzheimer's Disease. *Cell.* Jul 25 2019;178(3):536-551.e14. doi:10.1016/j.cell.2019.05.056
25. Lucin KM, O'Brien CE, Bieri G, et al. Microglial beclin 1 regulates retromer trafficking and phagocytosis and is impaired in Alzheimer's disease. *Neuron.* Sep 4 2013;79(5):873-86. doi:10.1016/j.neuron.2013.06.046
26. Keller CW, Kotur MB, Mundt S, et al. CYBB/NOX2 in conventional DCs controls T cell encephalitogenicity during neuroinflammation. *Autophagy.* May 13 2020:1-15. doi:10.1080/15548627.2020.1756678
27. Oosterhof N, Chang IJ, Karimiani EG, et al. Homozygous Mutations in CSF1R Cause a Pediatric-Onset Leukoencephalopathy and Can Result in Congenital Absence of Microglia. *Am J Hum Genet.* May 2 2019;104(5):936-947. doi:10.1016/j.ajhg.2019.03.010
28. Wang H, Newton G, Wu L, et al. CD47 antibody blockade suppresses microglia-dependent phagocytosis and monocyte transition to macrophages, impairing recovery in EAE. *JCI Insight.* Nov 8 2021;6(21)doi:10.1172/jci.insight.148719

REVIEWERS' COMMENTS

Reviewer #1 (Remarks to the Author):

The authors have revised the manuscript and properly addressed all points raised by the reviewers.

Reviewer #2 (Remarks to the Author):

The authors addressed most of the reviewer's concerns and the modifications made including the new data have strengthened the manuscript and its conclusions. There are no further comments on the manuscript.

Reviewer #3 (Remarks to the Author):

I would first like to thank the authors for their efforts in answering to all the concerns raised by the reviewers and adding significant new data to the manuscript. However, I still do have some concerns/suggestions mainly, but not exclusively, related to the new data.

Although the overall data does support the existence of ADAM in aged animals, I would like the authors to clarify/improve the next points:

-line 77: "ERK1/2 induces autophagosome maturation". After checking the references provided by the authors, this relationship is not straightforward to me. Indeed, as I understand, ERK phosphorylation as well as dephosphorylation have been linked with the regulation of different steps of the autophagic response.

-I was not able to find which exact phosphorylations of ERK, AMPK and Akt were detected by specific antibodies in this work. This information should be included in the methods section. Building a table with all the antibodies used in the work could be an option.

-Figure 3A: I suggest the authors including representative zoom-in detail images of p-ERK high cells with more LC3 dots as well as p-ERK low cells with fewer LC3 dots. Both types of cells could be present in the same image and differentiate them with different types of arrows, for example. I also suggest the authors to show all the individual channels for the zoom-in detail images as well as the merged channel. The image panel shown currently does not represent the quantification in Figure 3C.

-Figure 4G: Although it is clear that ULK1^{fl/fl} mice contain less P2RY12, P-ERK and LC3 staining than WT mice, it would be nice to see representative detail zoom-in images again for all the individual channels as well as the merged channel for the WT mice. It would be useful again to differentiate between pERK high and low microglia at the image. I understand that these images belong to the gray matter; this information should appear at the legend of the figure. White matter and/or spinal cord images could be shown at supplemental figures to support the quantification shown in Figure 4H.

-Line 258: the authors indicate that the loss of ADAM microglia impacted at the cortical brain regions. I suggest the authors specifying which brain regions were analyzed. It is not clear to me if authors analyzed the cortex as representative of gray matter, which was the area analyzed as representative of white matter, and which level of the spinal cord was analyzed. I suggest the authors providing more information on this.

-Figure 6C panel does not correspond with reference of Figure 3C in the text (lines 301-303). Indeed, Figure 6C is showing Figures 6F and 6H at the text (lines 310-311).

-I was not able to find the Table S2 with the statistical analysis details. I think that Figure 5A and Figure 7C statistical analysis would be more correct if two-way ANOVA was used (factors: treatment x genotype). Microglia is missing in the legend text of Figure 7C.

-Line 374: "this changed phosphorylation pattern is highly indicative of increased canonical autophagy". I suggest the authors modifying this statement to "this changed phosphorylation pattern was associated with increased canonical autophagy", unless they can prove that indeed high levels of p-ERK are highly indicative of autophagy flux augmentation in different cellular contexts.

Point-by-point

REVIEWERS' COMMENTS

Reviewer #1 (Remarks to the Author):

The authors have revised the manuscript and properly addressed all points raised by the reviewers.

Reviewer #2 (Remarks to the Author):

The authors addressed most of the reviewer's concerns and the modifications made including the new data have strengthened the manuscript and its conclusions. There are no further comments on the manuscript.

Reviewer #3 (Remarks to the Author):

I would first like to thank the authors for their efforts in answering to all the concerns raised by the reviewers and adding significant new data to the manuscript.

However, I still do have some concerns/suggestions mainly, but not exclusively, related to the new data.

Although the overall data does support the existence of ADAM in aged animals, I would like the authors to clarify/improve the next points:

-line 77: "ERK1/2 induces autophagosome maturation". After checking the references provided by the authors, this relationship is not straightforward to me. Indeed, as I understand, ERK phosphorylation as well as dephosphorylation have been linked with the regulation of different steps of the autophagic response.

Response:

The relationship between the ERK pathway and autophagy is complex, showcasing reciprocal regulatory interactions. Our standpoint continues to emphasize the association of elevated ERK activity with heightened autophagy. Nevertheless, we acknowledge the need for a more comprehensive and nuanced perspective, as outlined in the revised statement below.

"While Akt and AMPK act in an opposing manner on autophagy initiation, activation of ERK1/2 is both regulated by and regulates autophagy and is e.g., shown to induce autophagosome maturation³⁵⁻³⁸."

-I was not able to find which exact phosphorylations of ERK, AMPK and Akt were detected by specific antibodies in this work. This information should be included in the methods section. Building a table with all the antibodies used in the work could be an option.

Response:

Thank you for highlighting this crucial matter. Information regarding phosphorylation targets can now be found not only in the Results section but has also been included in the Methods and the Technical Data File for comprehensive access.

Text in methods section:

“The phosphorylation targeted were: ERK1/2-Thr202/Thyr204, AMPK-Thr183/172 and Akt-Ser473”

-Figure 3A: I suggest the authors including representative zoom-in detail images of p-ERK high cells with more LC3 dots as well as p-ERK low cells with fewer LC3 dots. Both types of cells could be present in the same image and differentiate them with different types of arrows, for example. I also suggest the authors to show all the individual channels for the zoom-in detail images as well as the merged channel. The image panel shown currently does not represent the quantification in Figure 3C.

Response:

Thank you for bringing this matter to our attention. We have incorporated representative images in Figure 3B illustrating cells with both high and low P-ERK1/2 levels, along with a magnified representative image highlighting structures identified as autophagosomes. The individual channels for the zoomed-in images are now available in the supplementary figure (see below). The quantified data in Figure 3C encompasses the assessment of 1) P-ERK1/2 high cells and 2) LC3B+ structures within these cells, comparing them to P-ERK1/2 low microglia.

New supplementary figure 2

(A) Zoom of immunocytochemistry images from figure 3A.

-Figure 4G: Although it is clear that ULK1fl/fl mice contain less P2RY12, P-ERK and LC3 staining than WT mice, it would be nice to see representative detail zoom-in images again for all the individual channels as well as the merged channel for the WT mice. It would be useful again to differentiate between pERK high and low microglia at the image. I understand that these images belong to the gray matter; this information should appear at the legend of the figure. White matter and/or spinal cord images could be shown at supplemental figures to support the quantification shown in Figure 4H.

Response:

We acknowledge your request, and in response, we have included zoomed-in images for Figure 4G in Supplementary Figure 2. These images now feature individual channels and indicators denoting what we classify as P-ERK1/2^{High} cells. Additionally, images of the white matter region and spinal cord have been incorporated into Supplementary Figure 2. See images below.

New supplementary figure 3

(A) Zoom of immunocytochemistry images from figure 3A. Scale bar represent 50 μ m

(B) Immunohistochemistry images of microglia in brain cortical regions. Arrows indicate P-ERK1/2^{High} cells. Zoom of image from figure 4G. Scale bar represent 100 μm

(C) Immunohistochemistry images of microglia in white matter and spinal cord regions.

Quantification shown in figure 4H. Scale bars represent 100 μm in images of *Brain white matter* and 200 μm in images of *Spinal cord*

-Line 258: the authors indicate that the loss of ADAM microglia impacted at the cortical brain regions. I suggest the authors specifying which brain regions were analyzed. It is not clear to me if authors analyzed the cortex as representative of gray matter, which was the area analyzed as representative of white matter, and which level of the spinal cord was analyzed. I suggest the authors providing more information on this.

Response: We have now added information on this matter to the result section (see below)

“Immunohistochemical analysis revealed a notable decrease of the ADAM population, predominantly in the grey matter regions of the cerebral cortex (Fig. 4g, h, S2b). Within these regions, we observed an increased density of LC3 in the P-ERK1/2^{High} population in Wt mice, which may suggest the initiation of autophagosome formation (Fig. 4i). We did not detect a significant reduction of ADAM or the microglia population at large in cerebral periventricular white matter regions or spinal cord, here analyzed at thoracic level (Fig. 4g, h, S2c). “

-Figure 6C panel does not correspond with reference of Figure 3C in the text (lines 301-303). Indeed, Figure 6C is showing Figures 6F and 6H at the text (lines 310-311).

Response: Corrected in text. Thank you for bringing this to our attention.

-I was not able to find the Table S2 with the statistical analysis details. I think that Figure 5A and Figure 7C statistical analysis would be more correct if two-way ANOVA was used (factors: treatment x genotype).

Response:

Thank you for your suggestion. We have updated the statistical analysis to employ a two-way ANOVA in both Figure 5A and 7C. Additionally, Table S2, containing comprehensive statistics and data, is now included in the manuscript files

-Microglia is missing in the legend text of Figure 7C.

Response: Information added.

-Line 374: “this changed phosphorylation pattern is highly indicative of increased canonical autophagy”. I suggest the authors modifying this statement to “this changed phosphorylation pattern was associated with increased canonical autophagy”, unless they

can proof that indeed high levels of p-ERK are highly indicative of autophagy flux augmentation in different cellular contexts.

Response: We have changed the text to the suggested statement - “This changed phosphorylation pattern was associated with increased canonical autophagy”